# Application of Monitoring Network Design and Feedback Information for Adaptive Management of Coastal Groundwater Resources

**DOI:** 10.3390/ijerph16224365

**Published:** 2019-11-08

**Authors:** Alvin Lal, Bithin Datta

**Affiliations:** Discipline of Civil Engineering, College of Science & Engineering, James Cook University, Queensland 4811, Australia; bithin.datta@jcu.edu.au

**Keywords:** adaptive coastal aquifer management, saltwater intrusion, simulation-optimization, monitoring network design, feedback information, compliance monitoring

## Abstract

Optimal strategies for the management of coastal groundwater resources can be derived using coupled simulation-optimization based management models. However, the management strategy actually implemented on the field sometimes deviates from the recommended optimal strategy, resulting in field-level deviations. Monitoring these field-level deviations during actual implementation of the recommended optimal management strategy and sequentially updating the management model using the feedback information is an important step towards efficient adaptive management of coastal groundwater resources. In this study, a three-phase adaptive management framework for a coastal aquifer subjected to saltwater intrusion is applied and evaluated for a regional-scale coastal aquifer study area. The methodology adopted includes three sequential components. First, an optimal management strategy (consisting of groundwater extraction from production and barrier wells) is derived and implemented for optimal management of the aquifer. The implemented management strategy is obtained by solving a homogenous ensemble-based coupled simulation-optimization model. Second, a regional-scale optimal monitoring network is designed for the aquifer system considering possible user noncompliance of a recommended management strategy, and uncertainties in estimating aquifer parameters. A new monitoring network design objective function is formulated to ensure that candidate monitoring wells are placed in high risk (highly contaminated) locations. In addition, a new methodology is utilized to select candidate monitoring wells in areas representative of the entire model domain. Finally, feedback information in the form of measured concentrations obtained from the designed optimal monitoring wells is used to sequentially modify pumping strategies for future time periods in the management horizon. The developed adaptive management framework is evaluated by applying it to the Bonriki aquifer system located in Kiribati, which is a small developing island country in the South Pacific region. Overall, the results from this study suggest that the implemented adaptive management strategy has the potential to address important practical implementation issues arising due to noncompliance of an optimal management strategy and uncertain aquifer parameters.

## 1. Introduction

Groundwater in coastal aquifers is a source of freshwater for many communities residing near coastal zones. However, in many coastal areas, continuous unplanned extraction of groundwater has resulted in saltwater intrusion, which is regarded as a common environmental issue for groundwater-dependent coastal communities. Sustainable management of coastal groundwater resources is a key priority for many coastal communities, hydrologists, water resources planners, and government stakeholders. This study evaluates the application of a three-phase adaptive management framework for optimal and sustainable control of saltwater intrusion in coastal aquifers. In Phase 1, an optimal management strategy obtained by solving a coupled simulation-optimization (S/O) model is implemented for optimal management of the aquifer. The next phase (Phase 2) consists of the development of a regional-scale monitoring network for the aquifer system considering both user noncompliance of a recommended management strategy and uncertainties in estimating aquifer parameters. In the final phase (Phase 3), feedback information from the optimal monitoring wells (OMWs) is used to sequentially modify/update pumping strategies for future time periods in the management horizon.

Groundwater flow and transport models combined with optimization techniques have been widely used to develop water resources management strategies that meet a user-defined management objective [1]. A number of optimal coastal groundwater management strategies have been developed and verified using a coupled S/O methodology, which utilizes a groundwater numerical simulation model and an optimization model [2,3,4,5,6,7,8]. Coupling a variable-density flow and salt transport numerical simulation model (simulates aquifer responses to stresses) to an optimization model is found to be computationally demanding, requiring huge computational time. Hence, a trained approximation of the groundwater numerical simulation model termed as surrogate models (data-driven mathematical models) is extensively used in the coupled S/O models to prescribe computationally efficient management strategies. In hydrologic literature, the use of surrogate models to improve computational efficiency and feasibility of the S/O model is well-established. The alternative use of embedded linked S/O models with either the finite element or finite difference equations embedded as constraints for simulating transient three-dimensional density-dependent flow and transport processes in a coastal aquifer have proven to be computationally infeasible [9]. The computational complexity is further increased when multiple objective management problems are considered. The computational time required to generate a single solution on the Pareto front, even for a small study area by externally linking a numerical simulation model (FEMWATER) to an optimization model, took days of process time (central processing unit time) [6]. Therefore, the approach of utilizing trained surrogate models as reasonably accurate approximators of complex numerical models enhances the feasibility of identifying an optimal solution for regional-scale study areas. Common surrogate modeling tools used in coupled S/O models to prescribe optimal management strategies for saltwater intrusion control in coastal aquifers can be found in Sreekanth and Datta [10], Roy and Datta [11], and Lal and Datta [12,13,14]. Also, in some cases, ensemble surrogate models instead of a standalone surrogate model are used to develop robust management strategies considering aquifer parameters’ uncertainties and surrogate models’ prediction uncertainties. The application of ensemble surrogate-based coupled S/O models can be found in Sreekanth and Datta [15], Roy and Datta [16] and Lal and Datta [17].

The coupled S/O model provides optimal strategies for the management of coastal groundwater resources. However, the correct implementation of the recommended optimal management strategies on the field is always a concern for decision-makers. To monitor field-level deviations of a recommended management strategy due to uncertain aquifer parameters and user noncompliance, a well-designed, robust and efficient groundwater monitoring network design is essential. Major objectives, criteria, and procedures for designing reliable groundwater monitoring networks can be found in Yangxiao [18]. Some key reasons for formulating and developing a groundwater monitoring network includes groundwater level monitoring [19,20,21,22], contamination detection [23,24,25,26,27,28,29], groundwater quality assessment [30,31,32,33,34], and conflicting economical/financial factors [35,36,37]. In addition, a comprehensive review by Loaiciga et al. [38] has documented the most important approaches to consider when designing groundwater monitoring networks. In saltwater intrusion management projects, a properly designed groundwater monitoring network design helps in collecting groundwater quality data, during and after the implementation of an optimal management strategy. The collected field data can be used to assess and compare the compliance of an implemented management strategy with the targeted coastal aquifer management objectives.

An optimal monitoring network design is an important step towards the attainment of an adaptive coastal groundwater resource management goal. Adaptive management is crucial mainly to counteract issues arising from the field-level implementation of a recommended optimal management strategy. In an adaptive management structure, the management strategies for future time periods in the management horizon are sequentially updated using feedback information gathered from the OMWs. In the domain of saltwater intrusion research, only a few studies have developed and evaluated adaptive management methods for management of coastal groundwater resources. Recently, Sreekanth and Datta [39] developed an adaptive management methodology for saltwater intrusion control in coastal aquifers using an optimal monitoring network design and sequential modification of management strategies utilizing feedback information. Also, Dhar and Datta [40] developed and implemented a monitoring network design to monitor compliance of an implemented robust optimal aquifer management strategy. However, both of these previously developed methodologies used hypothetical/illustrative coastal aquifer systems. This study, however, uses a regional-scale island coastal aquifer system to assess the performance of the developed methodologies.

The application of a three-phase adaptive management framework for optimal and sustainable control of saltwater intrusion in a coastal aquifer system presented in this study is of great significance. This study illustrates the application of a combination of computational methodologies and an adaptive approach of utilizing feedback information for enhancing or ensuring the practicability of field implementation of a regional-scale natural resource management strategy. Adaptive management is intended to be an iterative cycle in which groundwater pumping strategies and policies are regularly revised/updated to changing pumping conditions and due to the uncertainties of aquifer parameters. The main goal of using an adaptive management strategy is to ensure the prescribed strategies and policies are accepted and correctly implemented onto the field. The implementation of these adaptive management strategies will ensure the correct execution of the prescribed policies and will suggest amendments to these pumping strategies if not correctly followed. The proposed approach emphasizes the practical aspects of implementing a realistic coastal aquifer management strategy especially considering the following two issues: First, the recommended management strategy for optimal temporal and spatial groundwater withdrawals may differ from what actual users implement as often as it may be very difficult to enforce the prescribed strategy. Second, even if the actually implemented strategy is identical to the optimal recommended withdrawal strategy, the impact on the aquifer may be different from predicted impacts due to ubiquitous uncertainties in the estimated and modeled parameters, aquifer boundary conditions, errors in measurements including those in initial conditions, and hydraulic heads. Therefore, the need arises to sequentially correct and modify the actually implemented strategy, based on feedback field measurement information from the sequentially designed and implemented monitoring network. Based on these field measurements, a new optimal management strategy is derived by again solving the optimal management model, with updated information (e.g., new hydraulic heads and saline concentration, which could be different from those earlier predicted). The new management strategy is a modified or updated version of the earlier obtained management strategy with its impacts differing from the predicted impacts. The revised management strategy to be implemented in the next sequence tries to modify the prescribed strategy to increase the possibility of matching the consequences or impacts with the original management goals. This approach also makes it possible to address the practical issues of deviations from prescribed optimal pumping strategies or errors in predicting the impacts of a prescribed strategy, even if exactly implemented. In addition, this approach also helps in the convergence of prescribed management goals by utilizing sequentially obtained feedback information in the form of sequential field measurements of salt concentrations to achieve the goals of management efficiently and effectively. The practical utility of this approach is enormous as this approach provides a solution to a very practical difficulty in making optimal coastal aquifer management strategies achieve its goals and objectives. This study, in particular, applies this approach to the Bonriki aquifer and evaluates its implications in terms of improving the effectiveness of sequentially using an integrated set of an optimal withdrawal strategy design model and a monitoring network design model to ensure desired outcomes in real life. Hence, this application, together with an evaluation of the performance of this integrated adaptive approach is certainly a novelty and of significance.

In addition, this study is a logical extension of the authors’ earlier work in which groundwater management methodologies were only tested using illustrative aquifer systems. The island aquifer (Bonriki aquifer) considered in this study is situated in Kiribati, which is a small developing island country in the central Pacific Ocean. This study was aimed to present a straightforward and step-wise methodology for adaptive management of the Bonriki aquifer system. The first-ever monitoring network design is formulated and implemented for adaptive management of the Bonriki aquifer. The Bonriki aquifer is a crucial life-sustaining resource for the local Kiribati community and needs sustainable coastal groundwater management strategies and policies. Hence, the development and application of the methodologies presented in this study is a significant contribution to the framework of sustainable water resources management and administration in developing countries of the Pacific Islands. The study also presents many methodological contributions. First, the study presents the combined use of a multi-objective optimization model, data clustering, and integer programming to achieve the targeted adaptive coastal groundwater management goal. Second, a relatively new technique, support vector machine regression (SVMR)-based prediction models are used in the coupled S/O model to prescribe optimal management strategies for the aquifer system. Third, the *k*-means clustering technique is used to obtain the locations of candidate monitoring wells. The *k*-means clustering technique is a distinctive clustering algorithm, which offers an efficient and simple method of data clustering [41]. A detailed explanation of the *k*-means clustering methodology is presented in Bandyopadhyay and Maulik [42] and Nazeer and Sebastian [43]. Fourth, a new monitoring network design objective function is formulated to ensure that optimal monitoring wells are located in high-risk areas (highly concentrated areas). Lastly, a recent study by Post et al. [44] concluded that more work focusing on the management options for groundwater pumping from the Bonriki aquifer and a reevaluation of the appropriate sustainable yield is necessary. This work, therefore, was designed to present a first-ever adaptive management framework for the Bonriki aquifer system, which if adopted, can be beneficial to the local South Tarawa community. The results and evaluations are new and represent an important step in the regional-scale application of adaptive management strategies for sustainable management of coastal aquifers.

## 2. Materials and Methods

Details of the methodology proposed for the three-phase adaptive coastal aquifer management framework are presented in the subsequent sections.

### 2.1. Phase 1: Prescription and Implementation of an Optimal Management Strategy

The first step towards adaptive management of coastal aquifers is the prescription and implementation of an optimal management strategy. In this study, an optimal management strategy was prescribed using a coupled S/O approach. To reduce computational time and to ensure feasibility, SVMR-based homogenous ensemble models were used as approximates of the simulation model in the S/O framework. Key details of the variable-density flow and salt transport numerical simulation model, homogenous SVMR ensemble models, and the management model are described below.

#### 2.1.1. Numerical Groundwater Flow and Transport Model

The FEMWATER computer code from licensed Groundwater Modeling Systems package was used to simulate saltwater intrusion processes in the coastal aquifer system. FEMWATER [45] provides a three-dimensional finite element-based methodology for simulating density-dependent flow (saturated and unsaturated) and transport processes in a coastal aquifer system. Successful application of various groundwater variable-density flow and salt transport numerical models developed using the FEMWATER code for solving coastal aquifer processes are evident in many studies, including Sreekanth and Datta [10], Roy and Datta [46], and Lal and Datta [47].

The principal flow equation is in the form of the modified Richards equation [45]:(1)$$\frac{\rho }{\rho^{{}^{\circ}}}F\frac{\partial h}{\partial t}=\nabla \cdot \left[K\cdot \left(\nabla h+\frac{\rho }{\rho^{{}^{\circ}}}\nabla z\right)\right]+\frac{\rho }{\rho^{*}}q$$where ρ is the water density at chemical concentration c, $\rho^{{}^{\circ}}$ is the referenced water density at zero chemical concentration, F is the storage coefficient, h is the pressure head, t represents time, ∇ is the del operator, K is the hydraulic conductivity, z is the potential head, $\rho^{*}$ is the density of injection water or the withdrawn water, and q is the volumetric flow rate of the source (recharge) and/or sink (pumping).

For the saltwater intrusion problem, the constitutive relationship between the fluid density and concentration takes the form,(2)$$\frac{\rho }{\rho^{{}^{\circ}}}=1+ɛ\frac{c}{c_{max}}$$where ɛ is the dimensionless density reference ratio, c is the material concentration in the queous phase, and $c_{max}$ refers to the maximum material concentration.

The governing transport equation defines the material transport through groundwater systems. These equations are derived based on the laws of continuity of mass and flux. Some key processes that are considered for explaining saltwater intrusion occurrences are advection, dispersion/diffusion, and injection/withdrawal. The transport equation [45] is given by
(3)$$\theta \frac{\partial c}{\partial t}+V\cdot \nabla C-\nabla \cdot \left(\theta D\cdot \nabla c\right)=-\left(\alpha^{\prime }\theta c\frac{\partial h}{\partial t}\right)+qc_{in}-\frac{\rho^{*}}{\rho }qc+\left(F\frac{\partial h}{\partial t}+\frac{\rho^{{}^{\circ}}}{\rho }V\cdot \nabla \left(\frac{\rho }{\rho^{{}^{\circ}}}\right)-\frac{\partial \theta }{\partial t}\right)c$$where θ refers to the moisture content, V is discharge, D is the dispersion tensor, $\alpha^{\prime }$ is the compressibility of the medium, and $c_{in}$ is the material concentration in the source. Equations (1) and (3) characterize the flow and transport processes in a coastal aquifer system, respectively. These two equations are coupled together by the density coupling coefficient, and by the Darcy velocities, which make the saltwater intrusion phenomenon extremely nonlinear. Hence, the finite element-based variable density flow and salt transport numerical simulation model were used to solve the flow and transport processes governing equations concurrently.

#### 2.1.2. Homogenous Support Vector Machine Regression-Based Ensemble Surrogate Models

Homogenous SVMR ensemble models were used as approximate simulators of the saltwater intrusion process in the S/O model. The support vector machine regression (SVMR) surrogate was used because it is a relatively new and popular supervised data-driven methodology for constructing surrogate models [48]. SVMR has been used in numerous recent prediction modeling studies. The newly developed SVMR-based surrogate models have been evaluated for efficiency and accuracy for hypothetical aquifer study areas, and the evaluation results were reported in Lal and Datta [12]. In addition, Lal and Datta [47] also established that SVMR prediction performance was relatively better than genetic programming-based surrogate models. The main focus of the present study was on monitoring network design and adaptive management of coastal aquifers using feedback information. Hence, a detailed description of the SVMR working principle is not presented here. To ensure the robustness of the optimal solutions, ensemble SVMR models were used to incorporate aquifer parameter uncertainty into the management model.

Hydraulic conductivity and porosity were the two uncertain aquifer parameters considered while developing the coastal aquifer management model. To train each SVMR model in the ensemble, paired sets of input (hydraulic conductivity, porosity, and random transient groundwater pumping patterns from active wells) and output (salinity concentration at salinity monitoring wells) datasets were generated. In this study, the aquifer materials within each layer were considered homogenous. The specific values of hydraulic conductivity and porosity for each homogenous layer were obtained from a specified log-normal distribution and normal distribution, respectively. Transient groundwater pumping patterns were generated using uniformly distributed Latin hypercube sampling (LHS) methodology. Different combinations of these two uncertain parameters for the respective aquifer layers were implemented into the variable density flow and salt transport numerical model, keeping the other parameters constant during the simulation period. Each set of pumping patterns were provided as inputs to the variable density flow and salt transport numerical model, with different combinations of the two uncertain parameter values yielding different output concentrations at each specified monitoring well. From each of the input–output datasets, 80% was used for training the SVMR models, while the remaining 20% was used for validating the performance of each SVMR model developed. The resulting validated SVMR standalone models were strategically combined into an ensemble model using the simple averaging methodology [49]. Each homogenous ensemble SVMR model was constructed for predicting salinity concentration at each monitoring well. The simple average methodology for constructing ensemble models is a popular technique and has been used in various research applications [50,51]. A mathematical expression for constructing an ensemble model (*E_n_*) by integrating the predicted outputs (*P*) of various standalone models (*n*) using the simple averaging methodology is given in Equation (4).
(4)$$E_{n}=\frac{1}{N}\overset{N}{\underset{n=1}{\sum }}P^{n}\;n=1,2,\ldots ,N$$
Once trained and validated, the prediction capabilities of the standalone SVMR and ensemble models were quantified using various statistical indices such as the root mean square error (RMSE), mean bias error (MBE), Pearson’s correlation coefficient (r), Nash–Schliffe efficiency (NSE), and index of agreement (IOA).

#### 2.1.3. Formulation of the Multi-Objective Coastal Aquifer Management Model

The main aim of the developed management model was to prescribe an optimal pumping strategy from the two groups of wells (production and barrier wells) and simultaneously maintain salinity concentration in the aquifer within specified permissible limits. Production wells (PWs) were designed to extract fresh groundwater for local consumption. The barrier wells (BWs) were installed closer to the sea-side boundary and were used to extract saline water. BWs acted as a form of hydraulic barrier, thus, preventing saltwater intrusion into the aquifer. The mathematical formulation of the two conflicting objectives considered in the management model is given below.

Objective 1: Maximization of the total groundwater pumping from the PWs.

(5)$$F_{1}\left(P_{\mathrm{PW}}\right)=\overset{N}{\underset{n=1}{\sum }}\overset{T}{\underset{t=1}{\sum }}{\mathrm{PW}}_{n}^{t}$$

Objective 2: Minimization of the total groundwater pumping from the BWs.
(6)$$F_{2}\left(P_{\mathrm{BW}}\right)=\overset{M}{\underset{m=1}{\sum }}\overset{T}{\underset{t=1}{\sum }}{\mathrm{BW}}_{m}^{t}$$
${\mathrm{PW}}_{n}^{t}$ represents pumping from n^th^ production well at time t, and ${\mathrm{BW}}_{m}^{t}$ denotes pumping from the m^th^ barrier well at time t. N, M, and T are the total numbers of PWs, BWs, and time steps specified in the management model, respectively.

The optimization problem also utilized various constraints and bounds. These constraints and bounds are described below.

Constraint set 1: Coupling of SVMR prediction models to the optimization model in terms of the pumping decision variables. This constraint links the SVMR surrogate model to the optimisation model so that it can evaluate the solutions presented by the optimizer.

(7)$$c_{i}=\xi \left(\mathrm{PWs},\mathrm{BWs}\right)$$

Constraint set 2: Specifying the permissible salinity concentrations at the respective monitoring wells.
(8)$$c_{i}\leq {c\;}_{\max,i}\;\forall i,t$$
$c_{i}$ represents salinity concentration at the *i^th^* monitoring well at the end of the management time horizon.

The upper and lower bounds on pumping from PWs and BWs are given by
(9)$${\mathrm{PW}}_{\min}\leq {\mathrm{PW}}_{n}^{t}\leq {\mathrm{PW}}_{\max}$$
(10)$${\mathrm{BW}}_{\min}\leq {\mathrm{BW}}_{m}^{t}\leq {\mathrm{BW}}_{\max}$$

### 2.2. Phase 2: Regional-Scale Monitoring Network Design

#### 2.2.1. Possible Deviations in Pumping and Aquifer Parameter Uncertainty

One of the key features of a monitoring network design is that it should be able to accommodate possible deviations in field-level implementation of the prescribed optimal pumping strategy and also uncertainties associated with the aquifer parameters. Therefore, in this study, uncertainty due to possible field-level deviations in implementation of a prescribed optimal pumping strategy, and uncertainty in aquifer parameter (hydraulic conductivity and porosity) estimates were incorporated in the design of an optimal monitoring network. First, to implement field-level deviation in the chosen optimal management pumping solution, slightly perturbed optimal pumping rates were utilized. To achieve this perturbation, a truncated normal distribution of the deviations, ranging from 0%–20% of the actual deviations in pumping rates, similar to Sreekanth and Datta [39] was considered. Second, uncertainty in aquifer parameter estimation was incorporated by utilizing different realization of the hydraulic conductivity and porosity values in the variable-density flow and salt transport numerical simulation model. Different realization of hydraulic conductivity and porosity were obtained using log-normal distribution and normal distribution, respectively. The perturbed input optimal pumping rates and different realization of the two uncertain aquifer parameters were used in the variable-density flow and salt transport numerical simulation model to obtain different realizations of salinity concentration at all candidate (potential) monitoring wells.

#### 2.2.2. Location of Candidate Monitoring Wells

Location of candidate monitoring wells demands careful consideration. A subset of these candidate monitoring wells will be selected as OMWs. Many times, only a certain portion of the model domain is used as locations of candidate monitoring wells. However, in real scenarios, any possible node on the model domain can be considered as a location of a candidate monitoring well. In this study, *k*-means clustering [52] was utilized to obtain locations of candidate monitoring wells representative of the entire model domain. Clustering of all existing nodes using the *k*-means clustering methodology ensured that candidate monitoring wells are chosen from the entire area of the model domain. The main idea of using *k*-means clustering is to categorize the set of nodes into *k* disjoint clusters, where *k* is fixed in advance. After convergence, the *k*-means clustering solution offers a centroid for each of the clusters. The node number closest to this centroid is chosen as the node, where a candidate monitoring well is to be installed.

#### 2.2.3. Formulation of the Optimal Monitoring Network Design Model

The main goal of designing an optimal monitoring network was to monitor compliance of the salinity concentration resulting from an optimal management strategy with actual concentrations. The concentration measurement information from a designed monitoring network also helps in obtaining feedback information on the actual impacts of the field-level implementation of a management strategy. This feedback information can be utilized to redesign the management strategy to better achieve the objectives. In many cases, monitoring data can be collected from numerous locations in the aquifer. However, this may be impractical and inefficient due to limited budget allocations for monitoring projects and due to redundancy in data collected [53]. Therefore, a key feature of a monitoring network design is to locate a permissible number of monitoring wells (within budgetary limit) at locations suitable for collecting useful and reliable monitoring data. For the present study, the average of the logarithmic concentration at each candidate monitoring well was maximized to ensure that candidate monitoring wells were placed in high-risk areas (highly concentrated areas). The objective function with respective constraints is described below.

(11)$$\mathrm{Maximize}\;\frac{\sum_{i=1}^{N}log(C_{i})}{N}Y_{i}$$(12)$$\mathrm{subject}\;\mathrm{to};\;\sum_{i=1}^{N}Y_{i}\leq M\;Y_{i}\;Є\;\left(0,1\right)$$where $C_{i}$ is the concentration at the $i^{th}$ candidate monitoring well and $Y_{i}$ is the decision variable indicating whether to install $\left(Y_{i}=1\right)$ or not to install $\left(Y_{i}=0\right)$ a monitoring well at the $i^{th}$ specified location. The symbol *M* represents the maximum number of monitoring wells permitted in the monitoring network (due to budgetary or other management limitations). Phase 2 is designed to obtain optimal monitoring well locations. As an adaptive measure, feedback information in the form of salinity concentration data will be used to sequentially modify the future year pumping rates. The monitoring network is designed once. However, the information from this network of wells is used sequentially for the modification of future year pumping rates.

### 2.3. Phase 3: Sequential Modification of the Management Strategy

Optimal coastal aquifer management strategies for sustainable control of saltwater intrusion are largely developed for long time horizons, *T*. However, they are implemented in smaller time steps, *t*. With the help of a properly designed monitoring network, it is possible to gather feedback information regarding the compliance and/or noncompliance of an implemented optimal management strategy, or noncompliance of the resulting concentrations with the predicted concentrations. This information can be utilized to sequentially modify and/or update the management strategy at succeeding time steps, improving the prospects of attaining saltwater intrusion management goals.

In this study, a management time horizon of 4 years (*T* = 4) was considered. However, a prescribed management strategy was implemented in steps of 1 year (i.e., *t* = 1, 2, 3, and 4). The implemented 4-year optimal pumping strategy (selected from the Pareto front) was obtained by solving the homogenous ensemble SVMR surrogate-based coupled S/O model. Yearly salinity concentration data at the designed optimal monitoring networks as a result of the implemented pumping rates can be obtained using the variable density flow and salt transport numerical model. However, in real field scenarios, it is a common practice that the prescribed optimal pumping strategy will not be accurately implemented, and/or, the actual concentrations resulting from an implemented strategy will differ from the predicted impacts due to various uncertainties in prediction. Here, for performance evaluation purposes only, the field-level deviations between actual and predicted concentrations after the first year of implementation (*t* = 1) of optimal pumping strategy is incorporated by simulating the concentrations taking into account random deviations of actual pumping rates from prescribed pumping rates. The perturbed pumping rates different from optimal prescribed pumping rates are generated by adding random errors (0%–20%) to the optimal prescribed pumping rates. The inclusion of these field-level deviations will affect the resulting salinity concentration and may lead to noncompliance with management goals in terms of permissible salinities. Again, for performance evaluation purposes, the actual salinity concentrations at the designed OMWs for each time step can be simulated for the perturbed pumping rates, using the variable density flow and salt transport numerical simulation model. In actual applications, no such artificial perturbation is required, as the actual concentrations will be measured in the field using the designed monitoring network. The optimal monitoring network designed to gather information about the noncompliance of the prescribed management strategy helps in updating or revising the management strategy for *t* = 2, 3, and 4 to achieve the original management goals. The multi-objective coupled S/O model was utilized to sequentially update the management strategy using feedback information from the earlier time steps. This was performed by re-running the S/O model for future time steps after updating the initial and boundary conditions acquired from the monitoring network. For sequential modification of the pumping rates using feedback information, the multi-objective management model is solved as a single objective optimization problem. Objective 2 (Equation (6)) of the original multi-objective management model (i.e., barrier well pumping for the selected management strategy) is added as an additional constraint. The other constraints (Equations (7) and (8)) and bounds (Equations (9) and (10)) of the optimization problem remained unchanged.

### 2.4. Case Study: Application and Evaluation of the Developed Methodology

The developed coastal aquifer adaptive management methodology was applied to the Bonriki aquifer located in Kiribati. Kiribati is a small Pacific Island developing country situated in the central Pacific Ocean. The Bonriki aquifer is located in South Tarawa, which is the most densely populated area in Kiribati. The geographic location of Kiribati and the Bonriki aquifer is presented in Figure 1. Extracted groundwater from the Bonriki aquifer is the main source of fresh water for the people of South Tarawa [54]. Approximately more than 60% of the South Tarawa population are dependent on the extracted groundwater from the Bonriki aquifer [55].

The Bonriki aquifer with an area of 1.50 km^2^ and a depth of 60 m was modeled using the FEMWATER computer package. The lithology data from Bosserelle et al. [56] was used to map the lithology of the Bonriki aquifer. Geologic data from 19 boreholes were utilized to characterize the Bonriki aquifer as a two-layer system. Layer 1 comprised of Holocene sediments and layer 2 consisted of Pleistocene sediments, similar to other small Pacific Island aquifers [57]. The Bonriki aquifer consisted of 19 PWs for the extraction of groundwater. Also, as a management option, 6 BWs were assumed to be located near the sea face (along the sea-side boundary) to hydraulically minimize saltwater encroachment into the aquifer. Six active monitoring wells (MWs) were also used for salinity concentration monitoring purposes. The entire model domain was horizontally discretized into a mesh comprising of small triangular elements. The model domain with specific well locations is presented in Figure 2.

The three boundaries surrounding the model domain are labeled as Boundary I, Boundary II, and the sea-side boundary. Boundary I and Boundary II are denoted with specified pressure head boundaries as heads along these two boundaries are not strictly zero. A pressure head of 1 m (at the top end) is assigned to Boundary I and II and allowed to vary linearly along the boundary until it reaches a constant value of 0 m at the sea-side boundary. The sea-side boundary is in direct contact with the sea. Hence, the sea-side boundary is specified as a constant head and constant concentration boundary. A constant head of zero and a constant concentration of 35,000 mg/L is assigned to this sea-side boundary. Groundwater recharge via rainfall was represented by a constant vertical flux across the entire model domain.

The field values of groundwater pumping rates, groundwater level (GL), and electrical conductivity (EC) data were obtained from Sinclair et al. [58]. During calibration and validation, pumping from all 19 PWs were considered. Barrier well pumping rate was set to zero. GL and EC data from 6 MWs were used for the calibration and validation process. The EC data were multiplied by a factor of 0.69 to obtain concentration data in mg/L, as proposed in Ghassemi et al. [59,60]. GL and EC data were available for a period of 17 months only (April 2013 to August 2014). The accessible 17 months of data were separated into two sets: SET I and SET II. SET I contained 12 months of data (April 2013 to March 2014), which were used for calibration. SET II contained 5 months of data (April 2014 to August 2014), which were used for validation.

The calibration process was performed using the trial and error approach. The initial hydraulic conductivity values for the two layers were obtained from Bosserelle, Jakovovic, Post, Rodriguez, Werner, and Sinclair [56]. Layer 1, which contained Holocene sediments, had very low hydraulic conductivity values when compared to Pleistocene sediments (layer 2). The large differences in the hydraulic conductivity of the two layers are documented in Bosserelle, Jakovovic, Post, Rodriguez, Werner, and Sinclair [56] and White et al. [61]. In this study, vertical heterogeneity of the two layers was considered with each having different representative hydraulic conductivities. Also, the hydraulic conductivity of both the layers was considered anisotropic in the x, y, and z directions. The average representative hydraulic conductivity of each layer was used to simplify the variable density flow and salt transport numerical model and also ensure its convergence. The other hydrological parameters of each layer were considered homogenous. The final representative hydraulic conductivity for both layers were obtained based on the calibration and validation process.

For calibration, the variable density flow and salt transport numerical simulation model was run in a transient mode (with monthly time step) for 334 days. After gradually and iteratively modifying hydraulic conductivity, porosity, and recharge within a reasonable range, an acceptable match (R^2^ value) between the field and simulated GL and concentration values at the 6 MWs were established. For calibration, the aim was to achieve a targeted R^2^ value >90%. The other parameters were obtained from available hydrologic literature (Table 1) and were omitted from the calibration process. The validation stage using SET II was initiated only after the R^2^ value >90% for all the 6 MWs was recorded. The calibrated aquifer parameters were verified by comparing them with the results from similar Pacific Island aquifer modeling studies (e.g., Underwood et al. [62]) and also with the results from the earlier developed SEAWAT model of the Bonriki aquifer presented in Bosserelle, Jakovovic, Post, Rodriguez, Werner, and Sinclair [56]. The final representative aquifer parameters used to develop the Bonriki aquifer model are listed in Table 1.

The validated model was used to evaluate the developed adaptive management coastal aquifer framework. Firstly, the multi-objective management model was evaluated using the developed variable-density flow and salt transport numerical model. All the 19 operational PWs and 6 BWs were used. A total management horizon of 4 years was considered. A total of 100 decision variables (25 wells × 4-year management horizon) were implemented into the S/O model. The maximum (1200 m^3^/day) and minimum (0 m^3^/day) pumping limits for both well types were added as bounds. The maximum allowable salinity concentration at the 6 MWs after the management period was incorporated as an optimization constraint. Each SVMR model was developed (trained and validated) using 700 input–output datasets. The 700 sets of randomized input pumping rates from both well types were generated using LHS methodology. The maximum (1200 m^3^/day) and minimum (0 m^3^/day) pumping limits for both the well types were added as bounds. Each set of the generated input pumping rates were fed to the variable-density numerical flow and transport model separately to obtain salinity concentration data at the respective monitoring wells. Input pumping and output concentration datasets were used to train and test an SVMR surrogate model for each location. The training and testing of the SVMR models in the ensemble were done on the MATLAB 2017a platform. To achieve satisfactory prediction results, the Gaussian kernel was used, with ε, *C*;, and γ (Gaussian kernel parameters) having a value of 0.60, 10, and 0.001, respectively. These parameter values were obtained after numerous experimental runs. Each standalone SVMR model could only predict the salinity concentration at a specific monitoring well. Ten different combinations of hydraulic conductivity and porosity values were used to develop 10 different SVMR models for each monitoring well. The prediction results of these 10 SVMR models were integrated into an ensemble. Thus, 6 ensemble SVMR models were developed to predict salinity concentration at the 6 corresponding MWs. For each SVMR model, the hydraulic conductivity input values were derived from a log-normal distribution with the calibrated value of hydraulic conductivity as the mean and using a variance of 0.4 m/d. Similarly, porosity input values were derived using a normal distribution with a calibrated value of porosity as the mean and a variance of 0.1.

The validated SVMR ensemble models were externally coupled to the multi-objective genetic algorithm (MOGA) optimization model using the MATLAB 2017a platform. One of the main reasons for using MOGA was its efficiency. In a single run, MOGA can provide the optimal Pareto front comprising of non-dominated solutions at the end of the stated number of generations. For MOGA implementation, a population size of 2000, function tolerance of 1 × 10^−4^, constraint tolerance of 1 × 10^−3^, Pareto front population fraction of 0.3, and crossover fraction of 0.8 were utilized. The number of generations used was fixed to 10,000. This value was obtained after trying different generation sizes for the convergence of the population. The constraint of the optimization (maximum allowable salinity levels at the 6 MWs labeled as MW1, MW2, MW3, MW4, MW5, and MW6) ensured that the salinity at the MWs was restricted to a pre-specified limit. The maximum tolerable salinities for MW1 and MW2 were set to 20,000 mg/L. MW1 and MW2 were closer to the shoreline and restricting the salinity levels at these locations to very lower levels was impractical. The maximum acceptable salinity concentration at MW3 and MW4 was set to 5000 mg/L and 4000 mg/L, respectively. Finally, the maximum acceptable salinity concentrations at MW5 and MW6 were set to 450 mg/L. MW5 and MW6 are located in an area of concentrated pumping well locations. It is anticipated that the water extracted from these locations are of good quality and suitable for consumption by the local South Tarawa communities.

For Phase 2, 100 candidate monitoring well locations were chosen using the *k*-means clustering methodology. The *k*-means clustering code was written and executed in the *R* platform. A fixed number of iterations was used as the stopping criteria. In the present case, 50 iterations were considered. Also, before the execution of the *k*-means clustering code, the number of candidate monitoring wells to be used in the monitoring network project were specified as the value of *k* (number of clusters). One hundred perturbed pumping rates from the chosen optimal pumping rates were obtained using the LHS strategy. These perturbed pumping rates and 100 different combinations of hydraulic conductivity and porosity were used in the variable-density flow and salt transport numerical simulation model to obtain 100 different realizations of salinity concentration at the 100 candidate monitoring wells. Ten (*M* = 10) optimal monitoring well locations out of the 100 candidate monitoring well locations were obtained by implementing the designed monitoring network. For Phase 3, the single objective optimization problem used for sequential modification of pumping rates for the future time periods was solved using the genetic algorithm optimization solver available in the MATLAB 2017 platform. A flow diagram of the proposed adaptive coastal aquifer management framework is presented in Figure 3.

## 3. Results and Discussions

### 3.1. Development and Execution of the Coupled S/O Model

#### 3.1.1. The Bonriki Aquifer Calibration and Validation Results

The calibration and validation results are presented in Figure 4 and Figure 5, respectively. As observed, the calibration and validation results signify the accuracy of the numerically simulated model in replicating saltwater intrusion processes in the Bonriki aquifer. The relative differences between the simulated and field groundwater level during the calibration and validation period was less than 10%. A similar trend was observed for the concentration data during the calibration and validation period. Despite limited datasets, the calibration and validation results showed that the developed variable-density flow and salt transport numerical model could approximate regional groundwater flow and transport characteristics of the Bonriki aquifer with reasonable precision.

#### 3.1.2. The Performance Evaluation of the Proposed Methodology Utilizing Homogenous Ensemble Models

The utility of using the integrated approach of management strategy development, implementation, and the subsequent modification based on feedback measurements obtained from a designed monitoring network were evaluated to establish the potential applicability for the selected aquifer site. The homogenous SVMR ensemble model was used to approximately simulate aquifer responses in the coupled S/O model consisting of 10 standalone SVMR models. Each standalone SVMR model was trained and tested using datasets obtained from different variable-density flow and salt transport numerical models developed using different combinations of hydraulic conductivity and porosity values. The predictive accuracy of each standalone model in the ensemble is shown in Table 2. It is observed that all the standalone models in the ensemble predicted salinity concentration at their respective monitoring wells with reasonable accuracy (quantified in terms of RMSE, MBE, r, NSE, and IOA). This accuracy of the standalone models eventually reflected on the accuracy of the ensemble models used in the coupled S/O model. The performances of the homogenous SVMR ensemble surrogate model is presented in Table 3.

#### 3.1.3. Implementation of the Optimal Aquifer Management Strategy

The executed homogenous SVMR ensemble-based coupled S/O model presented a Pareto front containing several trade-off, optimal solutions in a runtime of ~3 h. Each optimal solution on the Pareto front represents an optimal pumping strategy, which can be implemented as management policy. The total optimal pumping rates from all the PWs and all the BWs for the four year management horizon ranged from about 30,000–44,000 m^3^/day and 1000–9000 m^3^/day, respectively. These pumping rates obtained are based on the imposed constraints (specified permissible concentration limits placed at the different MWs) specified in the management model. The optimal pumping values were also within the specified bounds used in the optimization model. The maximum and minimum annual rainfall in Tarawa is approximately about 4300 mm and 2100 mm [56]. Based on this annual rainfall over a highly permeable aquifer top cover with the proportionately very small built-up area, it is reasonable to assume a vertical annual recharge rate of nearly 2000 mm. This vertical recharge amount is itself around 3 million m^3^ per year. Therefore, if the barrier well extraction rate is excluded from the total withdrawal computed above, as a large proportion of the barrier well extraction is contributed by the sea face constant head boundary, the total specified withdrawal from water supply wells nearly matches the vertical recharge estimated. Therefore, the recharge rate imposed appears to be reasonable.

Validation of these optimal solutions is a crucial step in S/O management framework. Validation of optimal solutions was carried out by randomly selecting a few optimal solutions from the Pareto front and implementing them on the original variable-density flow and salt transport numerical model. In this study, five random optimal solutions were implemented into each of the 10 variable-density flow and salt transport numerical models and 10 standalone SVMR surrogate models. The average of the concentration values from 10 variable-density flow and salt transport numerical models is compared with the homogenous SVMR ensemble surrogate models. These comparison results are presented in Table 4. The percentage relative error for this comparison is less than 5% at all the MWs. This establishes the fact that the homogenous ensemble SVMR surrogate model approximated the variable flow and salt transport model with reasonable accuracy. Also, it was observed that the concentration values converged to the upper limit of the set constraints (e.g., at MW1, the maximum allowable salt concentration in the optimization model was specified as 20,000 mg/L). In the comparison results presented in Table 4, it is seen that for all five selected optimal solutions, the concentration values converged to the specified upper limit (20,000 mg/L). A similar pattern was observed for all other MWs.

The main focus of this study was to develop a monitoring network design for the Bonriki aquifer. For this purpose, a randomly selected optimal solution (solution *k* in Figure 6) was selected and implemented as a coastal aquifer management strategy. The total production well and barrier well pumping for the selected optimal solution were 39,728.44 m^3^/day and 3630.89 m^3^/day, respectively. The specific pumping from each PW and BW for the selected management strategy is demonstrated using Figure 7.

### 3.2. Optimal Monitoring Wells

The location of candidate monitoring wells obtained using the *k*-means clustering methodology is presented in Figure 8. The clustering methodology ensured that the candidate monitoring wells were scattered over the entire model domain. The truncated optimal pumping patterns (to demonstrate field-level deviation) for the implemented management strategy and uncertain aquifer parameters were used to generate 100 salinity concentration realizations at 100 candidate monitoring wells. Out of the 100 candidate monitoring wells, only 10 were selected as the OMWs. The monitoring network optimization formulation using the LINGO 17 platform [64] presented 10 OMWs, which are presented in Figure 9.

The average of the logarithmic concentration at each candidate monitoring well was maximized to ensure that candidate monitoring wells were placed in high-risk areas (highly concentrated areas). As seen in Figure 9, the location of all the OMWs were closer to the sea-side boundary, where salinity concentration as a result of production well pumping was the highest. Also, well-spread OMWs are observed, which avoid redundancy of monitoring well installation. The objective function used in the design of a monitoring network is only one possible objective. Other objectives based on different study area management scenarios can also be considered. The locations of the optimal monitoring wells are dependent on the monitoring network design objective functions and will be different for a different monitoring network objective. For example, maximizing weighted average concentrations at candidate monitoring locations as an objective function will result in a different set of optimal monitoring wells compared to the one designed in this study. However, for this particular study, a simple objective function is used to highlight the other aspects of linked S/O and the use of sequential information to modify management strategies over time.

### 3.3. Modified Pumping Rates Using Feedback Information

The selected management strategy from the Pareto front was used to adaptively modify pumping solutions based on the deviations in salinity concentration data (i.e., the difference in salinity concentration after utilizing the recommended (optimal) and actually implemented management strategy). The selected recommended four year management strategy was modified based on the feedback formation obtained based on the preceding year implemented pumping rates. For the selected management strategy, the total production well pumping and barrier well pumping rates were 39,728.4 m^3^/day and 3630.9 m^3^/day, respectively. The resulting salinity concentrations obtained at the OMWs at the end of year 1 as a result of the recommended pumping strategy are given in Table 5 as Situation A of year 1. It is highly likely that the pumping rates from the recommended management strategy will not be exactly implemented in the field. To account for this field-level deviation, the production well and barrier well pumping rates from the recommended management strategy were perturbed within 0%–20% of the recommended rates. This perturbation reflects the actual pumping rates implemented onto the field. However, this step is only relevant for performance evaluation purposes. In actual field applications, the deviations from intended consequences will be measured at specified monitoring wells in the designed monitoring network. The salinity concentration at the OMWs at the end of year 1 as a result of this perturbed implemented pumping rates are given as situation B of year 1. It is observed from Table 5 that the deviations in the pumping rates of year 1 also lead to a slight deviation in the concentration values. Based on this salinity concentration, the pumping rates for year 2, 3, and 4 are modified by rerunning the coupled S/O model, while keeping the other management constraints unchanged. The modified PW pumping rates are given in Table 6. The modifications were only done to pumping rates of year 2, 3, and 4 after gathering feedback information after the implementation of the year 1 pumping rates. These three changes are obtained based on the fresh solution of the optimization model utilizing feedback information from the optimal monitoring wells. The first year pumping rates remained unchanged.

The implemented year 1 pumping rates (9822.6 m^3^/day) and the future modified year 2, 3, and 4 modified pumping rates (29,737.1 m^3^/day) were 39,559.7 m^3^/day. This was less than the pumping rates from the originally recommended management strategy. The salinity concentration due to the modified recommended pumping rates at the end of year 2 is given in Table 5 as situation A of year 2. Situation B of year 2 is the salinity concentration due to the implementation of the perturbed pumping rates. Similarly, situation A and B for year 3 and 4 in Table 5 represent the salinity concentration due to the modified and perturbed pumping rates, respectively. The pumping rates for the future (i.e., year 3 and 4) are modified using the same procedure as discussed above. The modified objective function value for year 3 was 39,444.7 m^3^/day, which was also less than the total pumping rates from the original recommended management strategy. With the adaptive management framework, the total of four years implemented pumping rates obtained were 39,431.0 m^3^/day, which was less than the pumping rates from the recommended management strategy (39,728.4 m^3^/day). This is intuitively justifiable, as the actually implemented strategy for the initial year was suboptimal.

The solution results presented in Table 5 and Table 6 demonstrate that optimal pumping solutions recommended from the S/O model will need modification because of the field-level deviations encountered during the implementation process, or noncompliance by the user. As observed, the feedback salinity measurement information can be utilized to modify pumping rates for the remaining future time periods of the management horizon. Therefore, a properly designed optimal monitoring network and feedback information are crucial for adaptive management of coastal groundwater resources.

In general, the evaluated adaptive management methodology for the Bonriki aquifer system presented in the study gives promising results. Solving the multi-objective management model prescribed a set of optimal solutions in the form of a Pareto front. The obtained solutions are validated to ensure that the constraints are satisfied. Obtaining an optimal solution, and then exactly implementing it in the field are two different but critical issues discussed in this study. An optimal solution can be obtained using all the computational powers in hand. The issue of user non-compliance due to incorrect implementation of a selected optimal solution is the main concern addressed. To monitor the possible effects of user non-compliance and to update the subsequent time period optimal solution in order to rectify the outcome, feedback information in the form of salinity concentration data is obtained from the OMWs. The subsequently modified yearly pumping strategies help in converging to the original management goals in spite of earlier deviations from the prescribed strategy. All the computational powers can be used to develop an optimal pumping strategy for the aquifer system, but the question of user non-compliance remains the same and not entirely a computational issue. In a practical situation, we cannot guarantee if the optimal solution will be correctly implemented. In such scenarios, the adaptive management framework presented in this study will be useful. Theoretically, it is “possible” to search for and then identify an optimal solution by enormous enumerations although, for complex large-scale problems, it is totally impractical. The main contribution of optimization is to develop and use a formal search technique which efficiently searches for an optimal solution that is almost impossible to identify by enumeration. When more than one objective is considered, this becomes more critical.

## 4. Conclusions

The main goal of this study was to develop an adaptive management strategy for the management of coastal groundwater resources. Specifically, this study demonstrated the use of an integrated approach of utilizing an optimal management strategy, designed optimal monitoring network, and feedback information for adaptive management of an island coastal aquifer system. In achieving the targeted management goal, optimal production well and barrier well pumping strategies were considered as options for sustainable control of saltwater intrusion in the Bonriki aquifer system in the South Pacific. Using this derived optimal strategy, OMWs are identified. A new monitoring objective function is developed to determine OMWs in high salinity concentrated areas. The resulting OMWs are then used to monitor the compliance of the recommended management strategy to those actually implemented in the field. Based on the field-level deviations between the actual and planned salinity levels, the pumping rates for future time periods in the management horizon are sequentially modified using the updated coupled S/O model. It is noted that field-level deviations during the implantation of the recommended pumping rates could lead to a significant difference in the salinity concentration at OMWs. Hence, updating the management model using the feedback information from earlier time periods can be crucial for the management of the Bonriki aquifer system. The results clearly establish that subsequently modified yearly pumping strategies help in achieving the original management goals in spite of earlier deviations from the prescribed strategy. The solution results presented in this study open pathways for further similar studies that could be undertaken in other small island countries, where saltwater intrusion due to excessive/disproportionate groundwater withdrawal is a threat to the sustainable beneficial use of freshwater resources. The developed and evaluated adaptive management methodology has the potential to be applied to other regional-scale coastal aquifers subjected to saltwater intrusion. However, this application would require the development of a variable-density numerical groundwater flow and transport simulation model for the proposed study area. Development of a variable-density numerical groundwater simulation model would necessitate numerical modeling skills, other software/computational requirements, and groundwater (head and concentration) datasets. These datasets are not always readily available and may require rigorous field surveys/investigations, which can be costly. One limitation of the present work is that not all aspects of various uncertainties relevant to such a regional-scale natural resource management study could be addressed. This was not within the scope of a single developmental study, and we keep these options for the future.

## Figures and Tables

**Figure 1 ijerph-16-04365-f001:**
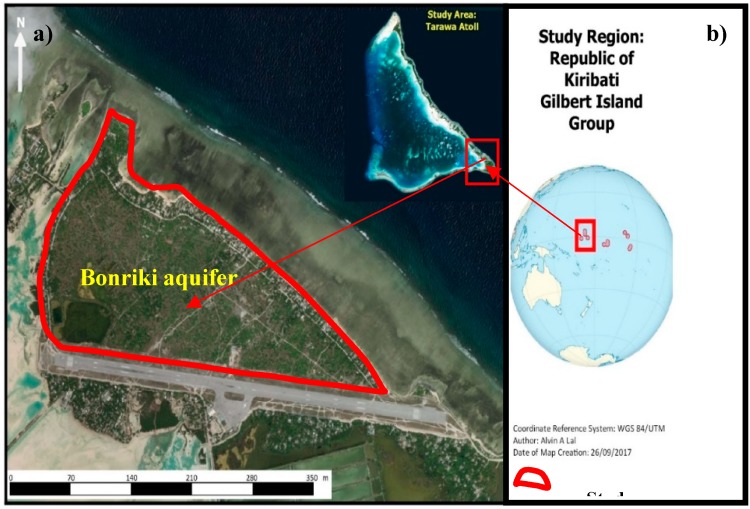
(**a**) Location of the Bonriki aquifer on Tarawa atoll and (**b**) geographical location of Kiribati in the central Pacific Ocean.

**Figure 2 ijerph-16-04365-f002:**
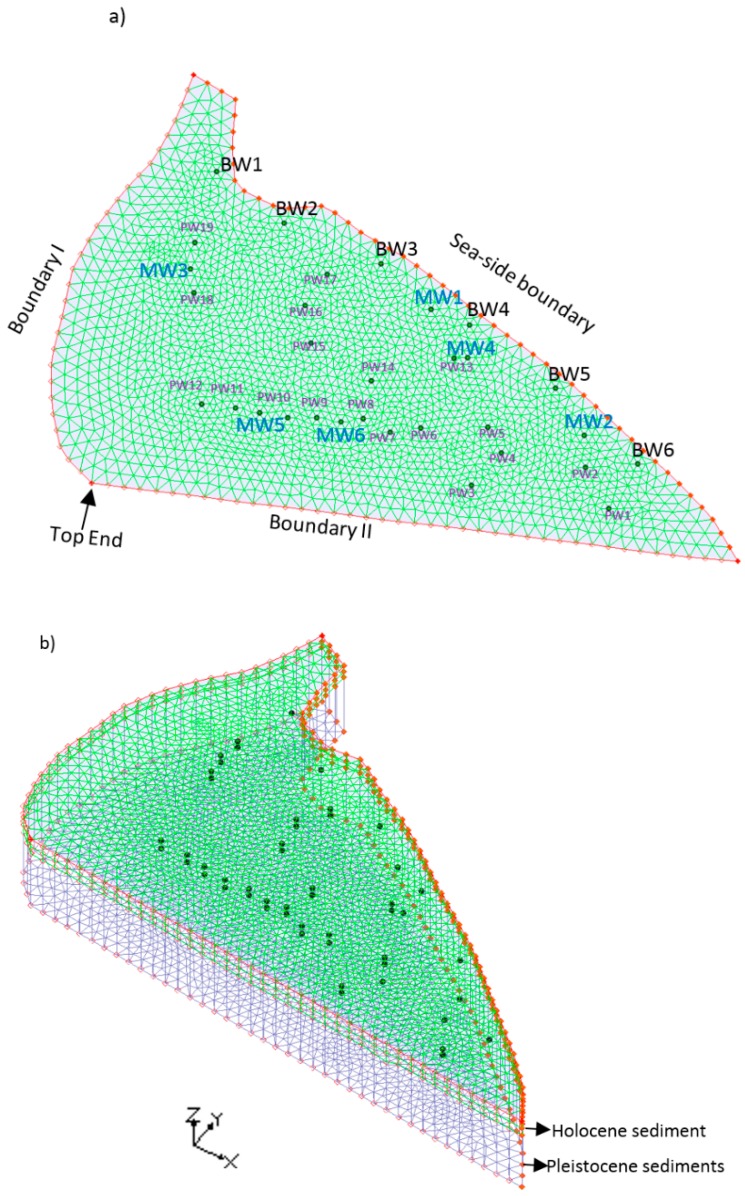
(**a**) Study area (Bonriki aquifer) model boundaries with specific well locations and (**b**) developed three-dimensional (3D) numerical simulation model.

**Figure 3 ijerph-16-04365-f003:**
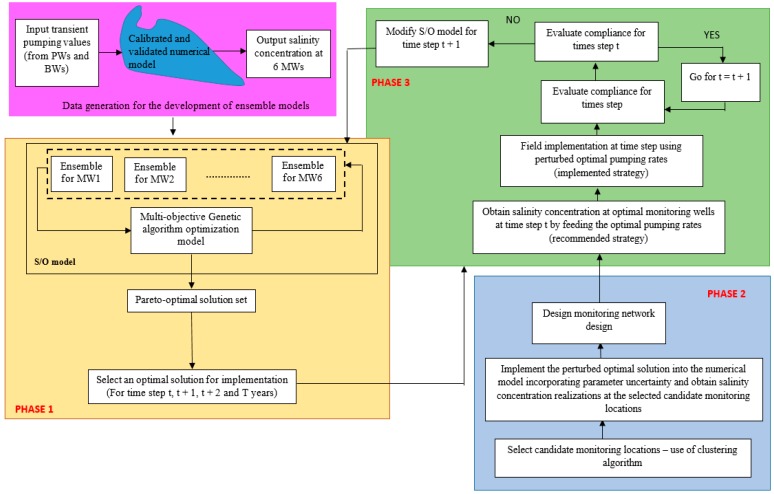
Flow diagram of the proposed three-phase adaptive management framework. Abbreviations: PWs: production wells; BWs: barrier wells; MW: monitoring well; S/O: simulation-optimization.

**Figure 4 ijerph-16-04365-f004:**
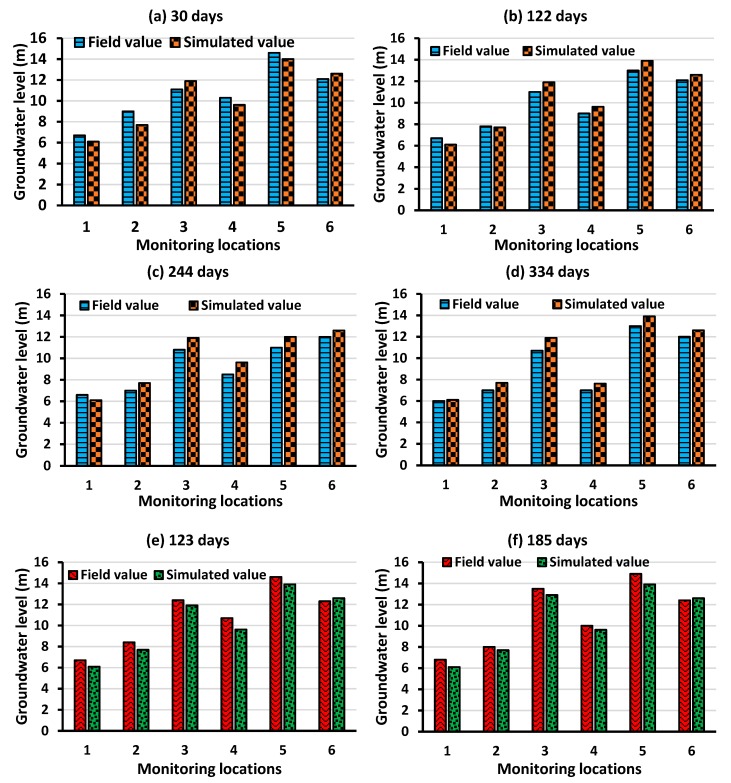
Field and simulated groundwater level values during calibration at (**a**) 30 days, (**b**), 122 days, (**c**) 244 days, and (**d**) 334 days; and validation period at (**e**) 123 days and (**f**) 185 days.

**Figure 5 ijerph-16-04365-f005:**
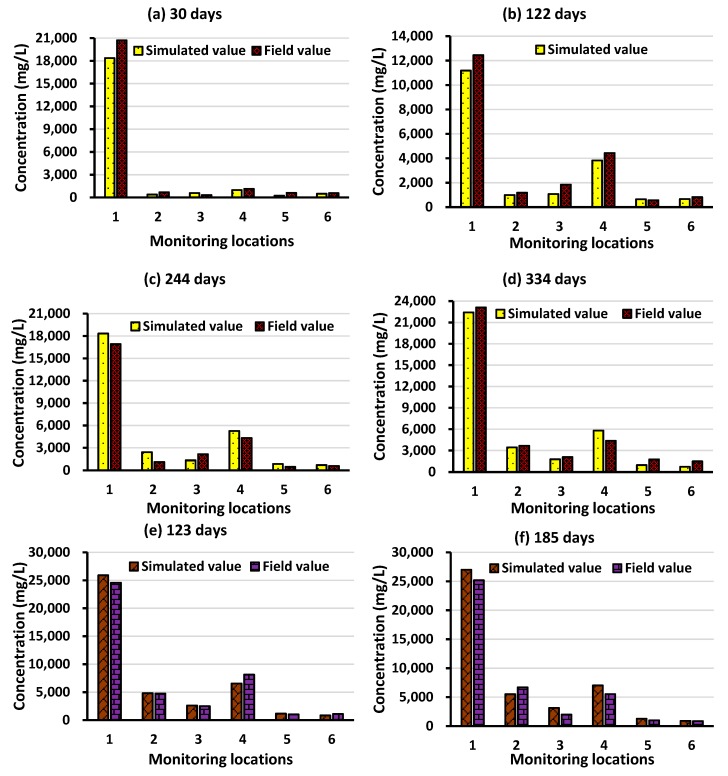
Field and simulated groundwater level values during calibration at (**a**) 30 days, (**b**), 122 days, (**c**) 244 days, and (**d**) 334 days; and validation period at (**e**) 123 days and (**f**) 185 days.

**Figure 6 ijerph-16-04365-f006:**
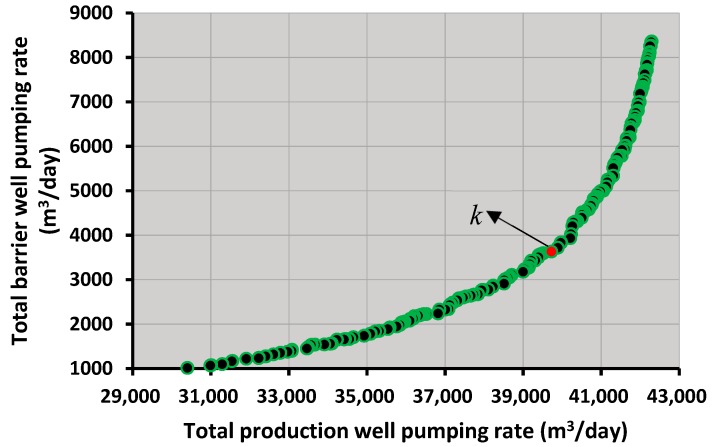
Pareto front displaying various trade-off optimal solutions.

**Figure 7 ijerph-16-04365-f007:**
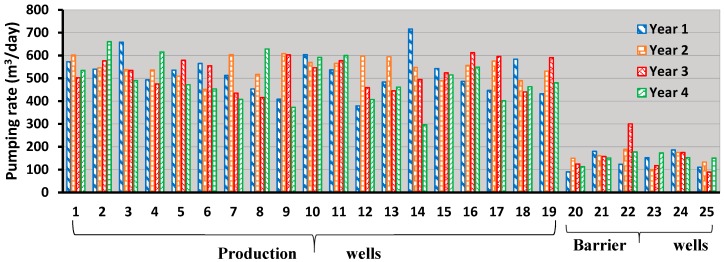
Specific pumping rates for the implemented management strategy.

**Figure 8 ijerph-16-04365-f008:**
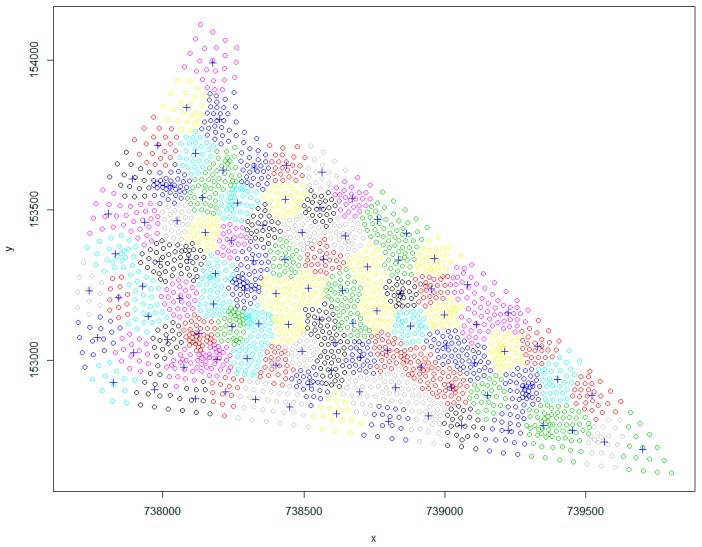
Location of the 100 candidate monitoring wells (**+**).

**Figure 9 ijerph-16-04365-f009:**
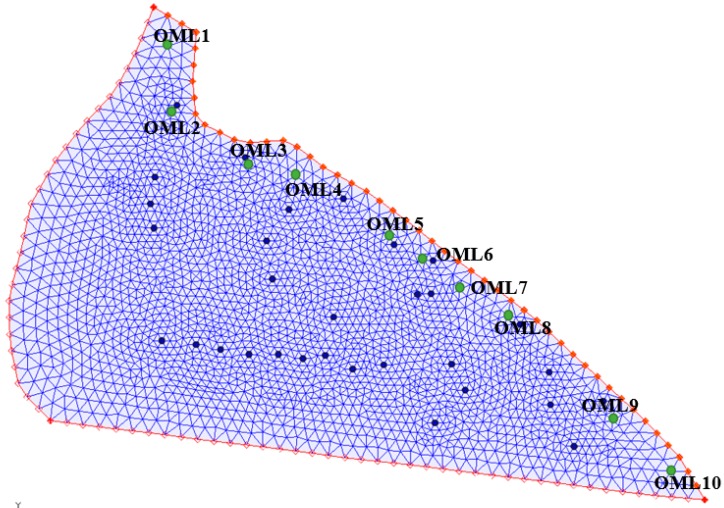
Locations of the 10 optimal monitoring wells (OMWs).

**Table 1 ijerph-16-04365-t001:** Bonriki aquifer parameter values.

Parameter		Values		Source
		Layer 1	Layer 2	
Hydraulic conductivity (m/day)	x	15	450	calibrated
	y	7.5	225	
	z	1.5	45	
Porosity		0.2	0.3	calibrated
Recharge		0.0055		calibrated
Seawater density (kg/m^3^)		1025		Oberdorfer et al. [63]
Freshwater density (kg/m^3^)		1000		Oberdorfer, Hogan, and Buddemeier [63]
Molecular diffusivity (m^2^/s)		1.5 × 10^−9^		Ghassemi, Jakeman, Jacobson, and Howard [60]
Dynamic viscosity of water (kg/ms)		280,985.8		-
Longitudinal dispersivity (m)		1		Bosserelle, Jakovovic, Post, Rodriguez, Werner, and Sinclair [56]
Lateral dispersivity (m)		0.05		Bosserelle, Jakovovic, Post, Rodriguez, Werner, and Sinclair [56]
Compressibility of water (m^2^/N)		4.4 × 10^−10^		Oberdorfer, Hogan, and Buddemeier [63]

**Table 2 ijerph-16-04365-t002:** Performance evaluation results of the standalone support vector machine regression (SVMR) models in the validation period. Abbreviations: RMSE: root mean square error; MBE: mean bias error; r: Pearson’s correlation coefficient; NSE: Nash–Schliffe efficiency; IOA: index of agreement.

Model	Evaluation Criteria	SVMR_1_	SVMR_2_	SVMR_3_	SVMR_4_	SVMR_5_	SVMR_6_
NM1	RMSE	5.10	6.17	3.74	2.95	2.02	1.89
	MBE	0.41	0.45	0.38	0.41	0.39	0.35
	r	0.96	0.97	0.97	0.97	0.98	0.98
	NSE	0.97	0.96	0.98	0.97	0.98	0.98
	IOA	0.94	0.95	0.95	0.96	0.96	0.96
NM2	RMSE	5.98	5.62	2.82	2.04	1.59	1.33
	MBE	0.47	0.56	0.62	0.52	0.39	0.44
	r	0.97	0.97	0.98	0.97	0.98	0.98
	NSE	0.96	0.96	0.96	0.97	0.97	0.97
	IOA	0.95	0.94	0.95	0.96	0.96	0.96
NM3	RMSE	4.16	5.22	3.51	4.86	3.02	2.14
	MBE	0.71	0.43	0.48	0.47	0.38	0.31
	r	0.97	0.96	0.97	0.96	0.98	0.98
	NSE	0.97	0.97	0.98	0.97	0.98	0.99
	IOA	0.94	0.95	0.95	0.94	0.95	0.96
NM4	RMSE	6.60	5.33	5.27	4.65	3.53	3.05
	MBE	0.52	0.55	0.64	0.64	0.43	0.48
	r	0.97	0.98	0.96	0.97	0.97	0.97
	NSE	0.97	0.96	0.96	0.97	0.97	0.98
	IOA	0.95	0.94	0.93	0.95	0.95	0.96
NM5	RMSE	6.96	7.13	5.12	5.68	4.25	4.56
	MBE	0.59	0.63	0.72	0.52	0.33	0.36
	r	0.97	0.97	0.97	0.97	0.98	0.97
	NSE	0.97	0.98	0.98	0.97	0.98	0.97
	IOA	0.95	0.94	0.95	0.94	0.96	0.95
NM6	RMSE	7.63	5.32	5.24	5.69	4.25	3.57
	MBE	0.44	0.65	0.66	0.46	0.41	0.34
	r	0.97	0.98	0.98	0.97	0.99	0.99
	NSE	0.98	0.98	0.98	0.98	0.98	0.99
	IOA	0.96	0.97	0.97	0.97	0.98	0.98
NM7	RMSE	7.26	6.75	6.03	5.87	5.66	5.12
	MBE	0.58	0.62	0.55	0.47	0.44	0.36
	r	0.97	0.98	0.98	0.98	0.98	0.99
	NSE	0.97	0.98	0.98	0.98	0.98	0.99
	IOA	0.96	0.97	0.97	0.97	0.98	0.98
NM8	RMSE	6.35	7.16	5.57	5.31	5.26	5.19
	MBE	0.54	0.58	0.52	0.49	0.33	0.41
	r	0.98	0.97	0.98	0.98	0.98	0.98
	NSE	0.97	0.97	0.97	0.97	0.97	0.97
	IOA	0.96	0.95	0.96	0.96	0.96	0.96
NM9	RMSE	7.37	6.89	8.43	6.22	5.32	4.41
	MBE	0.59	0.63	0.67	0.56	0.42	0.38
	r	0.98	0.98	0.96	0.97	0.97	0.98
	NSE	0.97	0.97	0.96	0.97	0.97	0.98
	IOA	0.95	0.96	0.95	0.96	0.97	0.97
NM10	RMSE	7.14	6.59	6.91	5.88	4.71	4.28
	MBE	0.55	0.62	0.52	0.55	0.39	0.44
	r	0.98	0.98	0.98	0.98	0.98	0.99
	NSE	0.98	0.98	0.98	0.98	0.98	0.99
	IOA	0.96	0.97	0.97	0.97	0.97	0.98

**Table 3 ijerph-16-04365-t003:** Performance evaluation results of the homogenous ensemble models.

Evaluation Criteria	En_SVMR_1_	En_SVMR_2_	En_SVMR_3_	En_SVMR_4_	En_SVMR_5_	En_SVMR_6_
RMSE	4.70	5.61	3.34	2.99	2.16	1.79
MBE	0.42	0.44	0.36	0.41	0.33	0.31
r	0.97	0.97	0.98	0.97	0.98	0.98
NSE	0.98	0.97	0.98	0.98	0.98	0.98
IOA	0.96	0.96	0.96	0.97	0.97	0.97

En_SVMR_n_: homogenous ensemble surrogate model developed using 10 standalone SVMR surrogate models for monitoring well n.

**Table 4 ijerph-16-04365-t004:** Optimal solution validation results.

Solution Number	MW1		MW2		MW3		MW4		MW5		MW6	
	NM_av_ (mg/L)	En_SVMR_1_ (mg/L)	NM_av_ (mg/L)	En_SVMR_2_ (mg/L)	NM_av_ (mg/L)	En_SVMR_3_ (mg/L)	NM_av_ (mg/L)	En_SVMR_4_ (mg/L)	NM_av_ (mg/L)	En_SVMR_5_ (mg/L)	NM_av_ (mg/L)	En_SVMR_6_ (mg/L)
1	19,870.4	19,677.0	19,795.63	19,742.53	4868.18	4844.85	3963.43	3932.23	447.82	444.94	432.72	429.88
2	19,708.8	19,621.2	19,783.73	19,735.64	4811.80	4776.57	3959.52	3916.66	433.94	429.27	435.97	433.33
3	20,009.2	19,846.9	19,975.89	19,949.89	4931.85	4928.47	3944.47	3911.19	444.84	437.12	436.18	431.45
4	19,798.4	19,660.5	19,793.16	19,757.80	4915.77	4906.76	3868.29	3868.82	433.54	427.58	439.93	436.34
5	19,727.4	19,567.6	19,829.44	19,795.19	4828.40	4839.35	3972.34	3967.71	430.35	427.29	427.90	426.47

NM_AV_: average concentration values from all 10 variable density flow and salt transport numerical models.

**Table 5 ijerph-16-04365-t005:** Salinity concentration (mg/L) at the optimal monitoring wells.

	Year 1		Year 2		Year 3		Year 4	
OMW	Situation A	Situation B	Situation A	Situation B	Situation A	Situation B	Situation A	Situation B
**1**	24,168.1	24,135.2	25,951.3	25,612.3	27,952.7	27,956.3	30,215.3	30,258.1
**2**	23,256.9	23,247.7	24,696.1	24,616.6	26,151.9	26,146.5	27,298.6	27,204.7
**3**	23,055.8	23,016.3	24,856.2	24,843.2	25,871.4	25,886.9	26,598.3	26,577.2
**4**	24,136.8	24,089.6	24,623.6	24,647.9	25,027.0	25,049.7	26,884.3	26,813.6
**5**	17,452.3	17,486.3	17,898.2	17,954.6	19,560.1	19,587.8	22,389.7	22,384.0
**6**	19,585.6	19,546.2	20,115.0	20,168.8	22,895.4	22,905.7	25,468.9	25,424.0
**7**	23,657.0	23,641.3	24,891.3	24,923.6	26,454.7	26,484.7	28,355.8	28,397.1
**8**	24,556.3	24,587.3	25,831.3	25,838.5	27,206.8	27,198.2	28,114.0	28,046.8
**9**	23,584.0	23,547.0	24,669.3	24,646.9	26,158.3	26,144.3	27,138.2	27,138.2
**10**	25,136.6	25,136.2	26,882.2	26,876.3	28,654.2	28,679.3	30,219.0	30,158.5

Situation A: due to the recommended strategy; Situation B: due to implemented strategy; OMW: optimal monitoring well.

**Table 6 ijerph-16-04365-t006:** Modified production well pumping rates (m^3^/day) during the entire management horizon.

	Year 1		Year 2		Year 3		Year 4	
	R	I	R	I	R	I	R	I
Year 1	9946.9	9822.6						
Year 2	10,426.5		10,335.1	10,378.8				
Year 3	9957.6		9987.1		9874.2	9904.4		
Year 4	9397.5		9414.9		9369.2		9325.3	-
Total	39,728.4		29,737.1		19,243.4		9325.3	

R: recommended; I: implemented.

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
