# Peer review of "Application of Monitoring Network Design and Feedback Information for Adaptive Management of Coastal Groundwater Resources"

_ijerph, 2019, doi:10.3390/ijerph16224365_

Round 1

Reviewer 1 Report

The authors applied SVMR to forecast the salinity concentration. The developed adaptive management framework is evaluated by applying it to the Bonriki aquifer system. The results showed that the implemented adaptive management strategy has the potential to address important practical implementation issues arousing due to noncompliance of an optimal management strategy and uncertain aquifer parameters.

However, the only one method of SVMR was used to predict salinity concentration in this paper, the majority of the outcomes are quite obvious, thus not very interesting for a reader. Apart from this main drawback, which should require an extremely deep revision of the manuscript, there are other main issues that the authors surely must resolve before the manuscript can be suitable for publication:

The author should be explained the data is normalized or non-normalized. So, the results of the normalized and non-normalized of input data should be analyzed. Please explain how to get or calculate the parameters of SVMR? Time-series data of the saltwater intrusion process seem to be quite strongly autocorrelated during most of the time. Statistical methods such as the autocorrelation function (ACF) and the partial autocorrelation function (PACF) are generally employed for selecting appropriate machine learning models. Can you use saltwater intrusion time series to plot the figures about the autocorrelation function (ACF) and the partial autocorrelation function (PACF), and explain the significance of vertical and horizontal coordinates about ACF and PACF? Do the authors ever think about accuracy evaluation of the salinity concentration forecasting? How to consider the uncertainty of input vectors response to salinity concentration? What the authors are trying to achieve, and finally what are the primary results to be beneficial for coastal groundwater resources management? It is not clear from the conclusions. The authors are asked to clearly state which is the novelty of the paper for which the paper deserves publication.

Author Response

Review Comments and Authors Response

Manuscript ID: ijerph-607473

Title: Application of monitoring network design and feedback information for adaptive management of coastal groundwater resources

Authors: Alvin Lal & Bithin Datta

The authors would like to thank the reviewer for his/her time and valuable constructive comments. We have carefully addressed all the review comments. The manuscript has been revised in response to the reviewer’s comments. The authors believe that the comments and suggestions were helpful in improving the quality of the manuscript and clarifying issues. We are submitting the revised manuscript for consideration as advised by the Assistant Editor. All the issues raised by the reviewer has been clarified in our detailed responses as listed below. Also, gist of these responses is now included in the revised manuscript.

Reviewer #1

English language and style

( ) Extensive editing of English language and style required
(x) Moderate English changes required
( ) English language and style are fine/minor spell check required
( ) I don't feel qualified to judge about the English language and style

Yes

Can be improved

Must be improved

Not applicable

Does the introduction provide sufficient background and include all relevant references?

( )

(x)

( )

( )

Is the research design appropriate?

( )

(x)

( )

( )

Are the methods adequately described?

( )

(x)

( )

( )

Are the results clearly presented?

( )

( )

(x)

( )

Are the conclusions supported by the results?

( )

(x)

( )

( )

Comments and Suggestions for Authors

The authors applied SVMR to forecast the salinity concentration. The developed adaptive management framework is evaluated by applying it to the Bonriki aquifer system. The results showed that the implemented adaptive management strategy has the potential to address important practical implementation issues arousing due to noncompliance of an optimal management strategy and uncertain aquifer parameters.

However, the only one method of SVMR was used to predict salinity concentration in this paper, the majority of the outcomes are quite obvious, thus not very interesting for a reader. Apart from this main drawback, which should require an extremely deep revision of the manuscript, there are other main issues that the authors surely must resolve before the manuscript can be suitable for publication:

The author should be explained the data is normalized or non-normalized. So, the results of the normalized and non-normalized of input data should be analyzed. Please explain how to get or calculate the parameters of SVMR? Time-series data of the saltwater intrusion process seem to be quite strongly autocorrelated during most of the time. Statistical methods such as the autocorrelation function (ACF) and the partial autocorrelation function (PACF) are generally employed for selecting appropriate machine learning models. Can you use saltwater intrusion time series to plot the figures about the autocorrelation function (ACF) and the partial autocorrelation function (PACF), and explain the significance of vertical and horizontal coordinates about ACF and PACF? Do the authors ever think about accuracy evaluation of the salinity concentration forecasting? How to consider the uncertainty of input vectors response to salinity concentration? What the authors are trying to achieve, and finally what are the primary results to be beneficial for coastal groundwater resources management? It is not clear from the conclusions. The authors are asked to clearly state which is the novelty of the paper for which the paper deserves publication.

Authors’ response

The authors thank the reviewer for pointing out some key issues. We have carefully addressed all the concerns of the reviewer.

The main concern of the reviewer was regarding the use of the support vector machine regression (SVMR) technique for predicting salinity concentration in the aquifer. The SVMR technique is a relatively new methodology, which has been rarely applied to saltwater intrusion prediction or, management studies. The SVMR models developed in this study are capable of approximating the responses of the complex numerical simulation model. It is computationally demanding and infeasible to link a complex saltwater intrusion numerical simulation to an optimization model. Therefore, in this study, the SVMR models are used as an approximate simulator and then linked to a multi-objective genetic algorithm optimization model to develop computationally feasible management strategies for the aquifer system. As shown in the results, SVMR models performed reasonably well. The performance evaluation results presented in Table 2 and 3 show that it can predict salinity concentration in the aquifer efficiently. The performances of SVMR models are not compared to other algorithms. This is because of various reasons. First, the authors have already compared and established that SVMR models are superior to well-establish prediction models such as those based on Genetic Programming. This information is included in the revised manuscript (line 229 to 237 on Page 5). Second, the main focus of the paper is to develop an adaptive management strategy for the aquifer system, and not on the prediction models. SVMR models are only used to approximate the responses of the complex numerical simulation model in the linked simulation-optimization (S/O) model. Third, the submitted manuscript deals with a real case study application. The manuscript has a detailed description of the study area, description of the adaptive methodology, results, analysis and conclusion. The length of the manuscript will significantly increase if we compare the performances of the SVMR models with other modelling algorithms.

The author should be explained the data is normalized or non-normalized. So, the results of the normalized and non-normalized of input data should be analyzed.

The SVMR models are used in the linked simulation-optimization (S/O) model to approximate the responses of the complex numerical simulation model. Therefore, the SVMR models were trained and validated using input-output datasets from the numerical simulation model. The input datasets are the pumping rates from different pumping wells (production and barrier wells) for different time periods. Each SVMR model was developed (trained and validated) using 700 input-output datasets. The 700 sets of randomized input pumping rates from both the well types were generated using Latin Hypercube Sampling strategy. The maximum (1200 m3/day) and minimum (0 m3/day) pumping limits for both the well types were incorporated as bounds. Each set of the generated input pumping rates were fed to the variable-density numerical flow and transport model separately to obtain salinity concentration data at respective monitoring wells. This was repeated 700 times to generate set pairs of input-output datasets. The authors feel that there was no need to normalize the datasets. The generated pairs were then used to train and validate the SVMR prediction models. Please refer to lines 518 to 529 on Page 11 and 12.

Please explain how to get or calculate the parameters of SVMR? Time-series data of the saltwater intrusion process seem to be quite strongly autocorrelated during most of the time. Statistical methods such as the autocorrelation function (ACF) and the partial autocorrelation function (PACF) are generally employed for selecting appropriate machine learning models. Can you use saltwater intrusion time series to plot the figures about the autocorrelation function (ACF) and the partial autocorrelation function (PACF), and explain the significance of vertical and horizontal coordinates about ACF and PACF?

The authors thank the reviewer for this comment. This reviewer may have overlooked the fact that the Artificial Intelligence-based learning models actually utilize not one but numerous realization of responses due to stresses i.e., pumping from production and barrier wells. These methods learn from feature extractions from the patterns implicit in these cause and effect relationship. Time series characteristics are different issues which may not be relevant here. The autocorrelation is not even the main issue, it the responses of the system that is used for learning.

The parameters of SVMR models were obtained after numerous experimental runs (refer to lines 531 to 533 on page 12). In this study, Gaussian kernel was used, with ɛ, ? and ɣ (Gaussian kernel parameters) having a value of 0.60, 10 and 0.001, respectively. The performance evaluation results in Table 2 and 3 show that the SVMR models prediction results were comparable to the numerical simulation model. Therefore, the parameter values selected were deemed appropriate and more in-depth analysis was not performed. However, the authors understand that the performance of the prediction model depends on the training/testing datasets and the model parameters.

Do the authors ever think about accuracy evaluation of the salinity concentration forecasting?

Yes. The authors did think about the accuracy of the SVMR prediction models. That was one of the main issues addressed here. Evaluation of the prediction model is an important step. In the present work, the SVMR models were assessed using various statistical indices such as root mean square error (RMSE), mean bias error (MBE), Pearson’s correlation coefficient (r), Nash-Sutcliffe efficiency (NSE) and index of agreement (IOA). These are standard indices, employed to assess the match between the corresponding outputs from the numerical simulation model and the prediction model (refer to lines 262 to 265 on Page 6). The prediction model assessment results of the standalone and ensemble SVMR models are presented in Table 2 and Table 3, respectively.

How to consider the uncertainty of input vectors response to salinity concentration?

Actually, the artificial intelligence-based learning process starts with randomizing the inputs and then utilizing the responses.

The pumping rates from each well types (production and barrier wells) were used as inputs to the numerical simulation model. The resulting salinity concentration at monitoring wells were recorded as outputs. Therefore, salinity is the response randomly generated corresponding to the randomized input patterns. These input-output patterns were used to train and test the SVMR prediction models. The randomized 700 sets of input pumping rates were generated using the Latin Hypercube Sampling strategy. The Latin Hypercube Sampling is a statistical method for generating a near-random sample of parameter values from a multidimensional distribution. In addition to this, the present study also incorporates uncertainties in aquifer parameter, and uncertainty due to user non-compliance while developing the adaptive management strategy. Please refer to lines 518 to 529 on page 11 and 12. However, we have now mentioned in the conclusion that not all aspects of relevant uncertainties were addressed within the scope of this limited study.

What the authors are trying to achieve, and finally what are the primary results to be beneficial for coastal groundwater resources management? It is not clear from the conclusions.

The conclusion is slightly modified to highlight primary results and consequential benefits to coastal groundwater management (refer to lines 983 to 1015 on Page 24).

Modified conclusion

The main goal of this study was to develop an adaptive management strategy for the management of coastal groundwater resources. Specifically, this study demonstrated the use of an integrated approach of utilizing an optimal management strategy, designed optimal monitoring network, and feedback information for adaptive management of an island coastal aquifer system. In achieving the targeted management goal, optimal production well and barrier well pumping strategy is considered as an option for sustainable control of saltwater intrusion in the Bonriki aquifer system in the South Pacific. Using this derived optimal strategy, OMWs are identified. A new monitoring objective function is developed to determine OMWs in high salinity concentrated areas. The resulting OMWs are then used to monitor the compliance of the recommended management strategy to those actually implemented in the field. Based on the field-level deviations between the actual and planned salinity levels, the pumping rates for future time periods in the management horizon are sequentially modified using the updated coupled S/O model. It is noted that field-level deviations during the implantation of the recommended pumping rates could lead to a significant difference in the salinity concentration at OMWs. Hence, updating the management model using the feedback information from earlier time periods can be crucial for the management of the Bonriki aquifer system. The results clearly establish that subsequently modified yearly pumping strategies helps in achieving the original management goals in spite of earlier deviations from the prescribed strategy. The solution results presented in this study opens pathways for further similar studies that could be undertaken in other small island countries, where saltwater intrusion due to excessive/disproportionate groundwater withdrawal is a threat to the sustainable beneficial use of freshwater resources. The developed and evaluated adaptive management methodology has the potential to be applied to other regional-scale coastal aquifers subjected to saltwater intrusion. However, this application would require the development of a variable-density numerical groundwater flow and transport simulation model for the proposed study area. Development of a variable-density numerical groundwater simulation model would necessitate numerical modelling skills, other software/computational requirements and groundwater (head and concentration) datasets. These datasets are not always readily available and may require rigorous field surveys/investigations, which can be costly. One limitation of the present work is that not all aspects of various uncertainties relevant to such a regional-scale natural resource management study could be addressed. This was not within the scope of a single developmental study, and we keep these options for the future.

The authors are asked to clearly state which is the novelty of the paper for which the paper deserves publication.

The novelty of the study is clearly stated in the revised manuscript. Paragraphs 5 and 6 mentions novel contribution of this work. The revised paragraphs are given below. Also, please refer to lines 113 to 181 on pages 3 and 4.

Modified paragraph 5 and 6

The application of a 3 phase adaptive management framework for optimal and sustainable control of saltwater intrusion in a coastal aquifer system presented in this study is of great significance. This study illustrates the application of a combination of computational methodologies and an adaptive approach of utilizing feedback information for enhancing or ensuring the practicability of field implementation of a regional-scale natural resource management strategy. Adaptive management is intended to be an iterative cycle in which groundwater pumping strategies and policies are regularly revised/updated to changing pumping conditions and due to aquifer parameters uncertainty. The main goal of using an adaptive management strategy is to ensure the prescribed strategies and policies are accepted and correctly implemented onto the field. The implementation of these adaptive management strategies will ensure the correct execution of the prescribed policies and will suggest amendments to this pumping strategies if not correctly followed. The proposed approach emphasizes the practical aspects of implementing a realistic coastal aquifer management strategy especially considering the following two issues: First, the recommended management strategy for optimal temporal and spatial groundwater withdrawals may differ from what actual users implement as often it may be very difficult to enforce the prescribed strategy. Second, even if the actually implemented strategy is identical to the optimal recommended withdrawal strategy, the impact on the aquifer may be different from predicted impacts due to ubiquitous uncertainties in the estimated and modelled parameters, aquifer boundary conditions, errors in measurements including those in initial conditions and hydraulic heads. Therefore, the need arises to sequentially correct and modify the actually implemented strategy, based on feedback field measurement information from the sequentially designed and implemented monitoring network. Based on these field measurements, a new optimal management strategy is derived by again solving the optimal management model, with updated information, e.g., new hydraulic heads and saline concentration (which could be different from those earlier predicted). The new management strategy is a modified or updated version of the earlier obtained management strategy with its impacts differing from the predicted impacts. The revised management strategy to be implemented in the next sequence tries to modify the prescribed strategy to increase the possibility of matching the consequences or impacts with the original management goals. This approach also makes it possible to address the practical issues of deviations from prescribed optimal pumping strategies or errors in predicting the impacts of a prescribed strategy, even if exactly implemented. In addition, this approach also helps in convergence to prescribed management goals by utilizing sequentially obtained feedback information in the form of sequential field measurements of salt concentrations to achieve the goals of management efficiently and effectively. The practical utility of this approach is enormous as this approach provides a solution to a very practical difficulty in making optimal coastal aquifer management strategies achieve its goals and objectives. This study, in particular, applies this approach to the Bonriki aquifer and evaluates its implications in terms of improving the effectiveness of sequentially using an integrated set of optimal withdrawal strategy design model and a monitoring network design model to ensure desired outcomes in real life. Hence, this application, together with an evaluation of the performance of this integrated adaptive approach is certainly a novelty and of significance.

In addition, this study is a logical extension of the authors earlier work in which the groundwater management methodologies were only tested using illustrative aquifer systems. The island aquifer (Bonriki aquifer) considered in this study is situated in Kiribati, which is a small developing island country in the central Pacific ocean. This study was aimed to present a straightforward and step-wise methodology for adaptive management of the Bonriki aquifer system. A first-ever monitoring network design is formulated and implemented for adaptive management of the Bonriki aquifer. The Bonriki aquifer is a crucial life-sustaining resource for the local Kiribati community and needs sustainable coastal groundwater management strategies and policies. Hence, the development and application of the methodologies presented in this study is a significant contribution to the framework of sustainable water resources management and administration in the Pacific island developing countries. The study also presents many methodological contributions. First, the study presents the combined use of a multi-objective optimization model, data clustering and integer programming to achieve the targeted adaptive coastal groundwater management goal. Second, a relatively new technique, support vector machine regression (SVMR) based prediction models are used in the coupled S/O model to prescribe optimal management strategies for the aquifer system. Third, the k-means clustering technique is used to obtain locations of candidate monitoring wells. The k-means clustering technique is a distinctive clustering algorithm, which offers an efficient and simple method of data clustering [1]. A detailed explanation of the k-means clustering methodology is presented in Bandyopadhyay and Maulik [2], and [3]. Fourth, a new monitoring network objective function is formulated to ensure that optimal monitoring wells are located in high-risk areas (highly concentrated areas). Lastly, a recent study by Post, et al. [4] concluded that more work focusing on the management options for groundwater pumping from the Bonriki aquifer and a re-evaluation of the appropriate sustainable yield is necessary. This work, therefore, was designed to present a first-ever adaptive management framework for the Bonriki aquifer system, which if adopted, can be beneficial to the local South Tarawa community. The results and evaluations are new and represent an important step in the regional-scale application of adaptive management strategies for sustainable management of coastal aquifers.

Žalik, K. R., An efficient k′-means clustering algorithm. Pattern Recognition Letters 2008, 29, (9), 1385-1391. Bandyopadhyay, S.; Maulik, U., An evolutionary technique based on K-means algorithm for optimal clustering in RN. Information Sciences 2002, 146, (1), 221-237. Nazeer, K. A.; Sebastian, M. In Improving the Accuracy and Efficiency of the k-means Clustering Algorithm, Proceedings of the world congress on engineering, 2009; 2009; pp 1-3. Post, V. E.; Bosserelle, A. L.; Galvis, S. C.; Sinclair, P. J.; Werner, A. D., On the resilience of small-island freshwater lenses: Evidence of the long-term impacts of groundwater abstraction on Bonriki Island, Kiribati. Journal of Hydrology 2018.

Reviewer 2 Report

Comments and Suggestions for Authors

The paper entitled “Application of monitoring network design and feedback information for adaptive management of costal ground water resources”, by Lal and Datta, propose a three-phase adaptive management framework for a coastal aquifer subjected to saltwater intrusion is applied and evaluated by applying to the Bonriki aquifer system located in Kiribati in South Pacific region. The novel contribution of this paper should be addressed more clearly. The results lack a comparison with the state of the art technique. However, the paper needs to be thoroughly revised before being further considered for publication in International Journal of Enviromental Research and Public Health.

GENERAL COMMENTS:

The abstract and introduction are well structured and appropriate. The introduction accurately present existing studies and the technical challenges addressed by the authors. Section 2 (Materials and Methods) are the major weakness of the paper. Some of the description of method should be put in the Section 1 (Introduction), for example, it is better to move the introduction of K-mean clustering in Section 2.2.2 to the Section 1 (Introduction).

SPECIFIC COMMENTS: (Not exhaustive)

Line 188, the refer “Lin et al., 1997” should be correct to the same template.

Line 218-224, the description of [44], [12] and [13] repeat several times, I suggest rewriting this paragraph.

Line 353 “4 year”->”4 years”

Line 661 “R-Recommended; I –Implemented” can be put in the caption of Table 6.

Author Response

Review #2 Comments and Authors Response

Manuscript ID: ijerph-607473

Title: Application of monitoring network design and feedback information for adaptive management of coastal groundwater resources

Authors: Alvin Lal & Bithin Datta

The authors would like to thank the reviewer for his/her time and valuable constructive comments. We have carefully addressed all the review comments. The manuscript has been revised in response to the reviewer’s comments. The authors believe that the comments and suggestions were helpful in improving the quality of the manuscript and clarifying issues. We are submitting the revised manuscript for consideration as advised by the Assistant Editor. All the issues raised by the reviewer has been clarified in our detailed responses as listed below. Also, gist of these responses is now included in the revised manuscript.

Reviewer #2

English language and style

( ) Extensive editing of English language and style required
( ) Moderate English changes required
(x) English language and style are fine/minor spell check required
( ) I don't feel qualified to judge about the English language and style

Yes

Can be improved

Must be improved

Not applicable

Does the introduction provide sufficient background and include all relevant references?

(x)

( )

( )

( )

Is the research design appropriate?

( )

(x)

( )

( )

Are the methods adequately described?

( )

(x)

( )

( )

Are the results clearly presented?

( )

(x)

( )

( )

Are the conclusions supported by the results?

( )

(x)

( )

( )

Comments and Suggestions for Authors

Comments and Suggestions for Authors

The paper entitled “Application of monitoring network design and feedback information for adaptive management of costal ground water resources”, by Lal and Datta, propose a three-phase adaptive management framework for a coastal aquifer subjected to saltwater intrusion is applied and evaluated by applying to the Bonriki aquifer system located in Kiribati in South Pacific region. The novel contribution of this paper should be addressed more clearly. The results lack a comparison with the state of the art technique. However, the paper needs to be thoroughly revised before being further considered for publication in International Journal of Enviromental Research and Public Health.

GENERAL COMMENTS:

The abstract and introduction are well structured and appropriate. The introduction accurately present existing studies and the technical challenges addressed by the authors. Section 2 (Materials and Methods) are the major weakness of the paper. Some of the description of method should be put in the Section 1 (Introduction), for example, it is better to move the introduction of K-mean clustering in Section 2.2.2 to the Section 1 (Introduction).

SPECIFIC COMMENTS: (Not exhaustive)

Line 188, the refer “Lin et al., 1997” should be correct to the same template.

Line 218-224, the description of [44], [12] and [13] repeat several times, I suggest rewriting this paragraph.

Line 353 “4 year”->”4 years”

Line 661 “R-Recommended; I –Implemented” can be put in the caption of Table 6.

Response to comments from Reviewer #2

The main concern of the reviewer was regarding the novelty of the study. The authors have clearly stated the novelty and new contributions in the revised manuscript (refer to lines 113 to 181 on Pages 3 and 4 of the revised manuscript). The novelty is also discussed in responses to reviewer #1 comments.

The results of the present study could not be compared with any other studies. This is mainly because no other studies have developed adaptive management strategies for island aquifer systems. The results presented in this study are new. These results are an important contribution to the field of environment management research.

As suggested by the review, the paper is revised according to the following comments:

Section 2 (Materials and Methods) are the major weakness of the paper. Some of the description of method should be put in the Section 1 (Introduction), for example, it is better to move the introduction of K-mean clustering in Section 2.2.2 to the Section 1 (Introduction).

Suggestions incorporated. The introduction of K-means clustering is presented in the introduction. Refer to lines 171 to 175 on page 4.

“….the k-means clustering technique is used to obtain locations of candidate monitoring wells. The k-means clustering technique is a distinctive clustering algorithm, which offers an efficient and simple method of data clustering [1]. A detailed explanation of the k-means clustering methodology is presented in Bandyopadhyay and Maulik [2], and [3]”.

Line 188, the refer “Lin et al., 1997” should be correct to the same template.

Suggestion incorporated. Refer to line 195 on page 4.

Line 218-224, the description of [44], [12] and [13] repeat several times, I suggest rewriting this paragraph.

Suggestion incorporated. The paragraph is modified. Please refer to lines 229 to 237 on Page 5.

“The newly developed SVMR based surrogate models have been evaluated for efficiency and accuracy for hypothetical aquifer study areas, and the evaluation results were reported in Lal and Datta [4]. In addition, Lal and Datta [5] have also established that SVMR prediction performance was relatively better than genetic programming based surrogate models. The main focus of the present study was on monitoring network design and adaptive management of coastal aquifers using feedback information. Hence, a detailed description of the SVMR working principle is not presented here. To ensure the robustness of the optimal solutions, ensemble SVMR models were used to incorporate aquifer parameter uncertainty into the management model.”

Line 353 “4 year”->”4 years”

Suggestion incorporated. Refer to line 361 on Page 8.

Line 661 “R-Recommended; I –Implemented” can be put in the caption of Table 6.

The authors would like to keep “R-Recommended; I –Implemented” in the footnote of Table 6. This is to maintain consistency with other Tables. Scientific Tables do contain footnotes and the authors believe that having a footnote in Table 6 does not diminish the quality of the paper.

Žalik, K. R., An efficient k′-means clustering algorithm. Pattern Recognition Letters 2008, 29, (9), 1385-1391. Bandyopadhyay, S.; Maulik, U., An evolutionary technique based on K-means algorithm for optimal clustering in RN. Information Sciences 2002, 146, (1), 221-237. Nazeer, K. A.; Sebastian, M. In Improving the Accuracy and Efficiency of the k-means Clustering Algorithm, Proceedings of the world congress on engineering, 2009; 2009; pp 1-3. Lal, A.; Datta, B., Modelling saltwater intrusion processes and development of a multi-objective strategy for management of coastal aquifers utilizing planned artificial freshwater recharge. Modeling Earth Systems and Environment 2017, 1-16. Lal, A.; Datta, B., Development and Implementation of Support Vector Machine Regression Surrogate Models for Predicting Groundwater Pumping-Induced Saltwater Intrusion into Coastal Aquifers. Water Resources Management 2018, 1-15.

Reviewer 3 Report

The authors in this paper presented an approach for the management of coastal groundwater using coupled simulation-optimization based management models. In this study, a three-phase adaptive management framework for a coastal aquifer is applied and evaluated in a small study area. Optimal management strategy, optimal monitoring network design and a feedback information system was used to develop an overall management approach for coastal aquifers.

Though the study is interesting and will be useful for the readers, authors have already published similar work using the same approach previously. The previous study by Sreekanth and Datta (2014) have used similar techniques to develop a saltwater management strategy in coastal aquifers. Apart from the new study area I am unable to find anything substantially new in the paper, therefore I recommend rejection.

Sreekanth, J., & Datta, B. (2013). Design of an optimal compliance monitoring network and feedback information for adaptive management of saltwater intrusion in coastal aquifers. Journal of Water Resources Planning and Management140(10), 04014026. 

Author Response

Review Comments and Authors Response

Manuscript ID: ijerph-607473

Title: Application of monitoring network design and feedback information for adaptive management of coastal groundwater resources

Authors: Alvin Lal & Bithin Datta

The authors would like to thank the reviewer for his/her time and valuable constructive comments. We have carefully addressed all the review comments. The manuscript has been revised in response to the reviewer’s comments. The authors believe that the comments and suggestions were helpful in improving the quality of the manuscript and clarifying issues. We are submitting the revised manuscript for consideration as advised by the Assistant Editor. All the issues raised by the reviewer has been clarified in our detailed responses as listed below. Also, gist of these responses is now included in the revised manuscript.

Reviewer #3

English language and style

( ) Extensive editing of English language and style required
( ) Moderate English changes required
(x) English language and style are fine/minor spell check required
( ) I don't feel qualified to judge about the English language and style

Yes

Can be improved

Must be improved

Not applicable

Does the introduction provide sufficient background and include all relevant references?

(x)

( )

( )

( )

Is the research design appropriate?

(x)

( )

( )

( )

Are the methods adequately described?

(x)

( )

( )

( )

Are the results clearly presented?

(x)

( )

( )

( )

Are the conclusions supported by the results?

(x)

( )

( )

( )

Comments and Suggestions for Authors

The authors in this paper presented an approach for the management of coastal groundwater using coupled simulation-optimization based management models. In this study, a three-phase adaptive management framework for a coastal aquifer is applied and evaluated in a small study area. Optimal management strategy, optimal monitoring network design and a feedback information system was used to develop an overall management approach for coastal aquifers.

Though the study is interesting and will be useful for the readers, authors have already published similar work using the same approach previously. The previous study by Sreekanth and Datta (2014) have used similar techniques to develop a saltwater management strategy in coastal aquifers. Apart from the new study area I am unable to find anything substantially new in the paper, therefore I recommend rejection.

Sreekanth, J., & Datta, B. (2013). Design of an optimal compliance monitoring network and feedback information for adaptive management of saltwater intrusion in coastal aquifers. Journal of Water Resources Planning and Management140(10), 04014026. 

Response to comments from Reviewer #3

We thank the reviewer for agreeing that the study presents an interesting work that will be of interest to potential readers. However, with due respect, we do not agree with the reviewer's comment regarding the similarity of this work to Sreekanth and Datta [1]. This study is completely different from Sreekanth and Datta [1] and presents new methods and findings. These details are as follows:

The main difference is the evaluation of the adaptive management strategies using a real case study based coastal aquifer system. The Bonriki aquifer in Kiribati is a major source of fresh water for the local community. However, the fresh groundwater in Bonriki is susceptible to saltwater intrusion due to excessive groundwater pumping. This study presents a first-ever adaptive management framework for the management of the Bonriki aquifer system. The numerical simulation model developed using the FEMWATER code is calibrated and validated using the available field data. This is another key contribution of the present study. The development of a saltwater intrusion numerical simulation model, calibration and validation, and its application is a new contribution, which will be of interest to potential journal readers. Sreekath and Datta [1] evaluated the method using an illustrative aquifer system, which did not require any calibration and validation. The use of SVMR prediction model instead of the complex numerical simulation model. Sreekanth and Datta [1] did not use the SVMR technique. As mentioned in response to comments by revivewer#1, SVMR is a relatively new technique and has been rarely used in saltwater intrusion management studies. The performance of SVMR models is found to be superior to other well-established models such as GP. Therefore, the utilization of SVMR models in developing adaptive management strategies for regional-scale coastal aquifer systems can deliver robust solutions. The use of clustering technique to obtain candidate monitoring locations. This is another new contribution. Sreekanth and Datta [1] randomly selected candidate monitoring locations. This is not logical as any node in the model domain can be used as a potential candidate monitoring locations. Therefore, in this study, the k-means clustering technique is used to obtain sets of candidate monitoring locations representative of the entire model domain. This is a more realistic approach when selecting candidate monitoring wells. This method has never been reported in any coastal aquifer management studies. The monitoring network design objective function used in this study is new and has not been used in any saltwater intrusion management studies. Sreekanth and Datta [1] used a different objective function, which focused on variance. In this study, we have used maximisation of the logarithmic mean concentration to ensure that optimal monitoring wells are placed ta high-risked (highly contaminated) areas. This is another new contribution of the present study.

Overall, this study illustrates the application of a combination of computational methodologies and an adaptive approach of utilizing feedback information for enhancing or ensuring the practicability of field implementation of a regional-scale natural resource management strategy. Also, this study presents several new methods and results, which has never been reported before. Therefore, the authors' believe that this study offers significant contributions, which will be beneficial to the interested audience. These details are presented in the introduction (paragraphs 5 and 6 – lines 113 to 181 on pages 3 and 4) and the conclusion (lines 837 to 869 on page 24).

Sreekanth, J.; Datta, B., Design of an optimal compliance monitoring network and feedback information for adaptive management of saltwater intrusion in coastal aquifers. Journal of Water Resources Planning and Management 2013, 140, (10), 04014026.

Round 2

Reviewer 1 Report

Paper has been revised adequately, hence it's now in an acceptable stage.

Reviewer 2 Report

The paper entitled “Application of monitoring network design and feedback information for adaptive management of costal ground water resources”, by Lal and Datta, propose a three-phase adaptive management framework for a coastal aquifer subjected to saltwater intrusion is applied and evaluated by applying to the Bonriki aquifer system located in Kiribati in South Pacific region.

The authors improved this manuscript acoording to the review report, some of the key issues are addressed propoerly. So I agree to accept this paper after minor revision (corrections to minor methodological errors and text editing).

Reviewer 3 Report

  I am happy with the response and recommend publication of the revised manuscript.